# Human and mouse neutrophils share core transcriptional programs in both homeostatic and inflamed contexts

Nicolaj S. Hackert [1,2,3,4,12], Felix A. Radtke [1,2,5,6,7,12], Tarik Exner[1,2,12], Hanns-Martin Lorenz[1], Carsten Müller-Tidow [8,9], Peter A. Nigrovic [3,5], Guido Wabnitz [2] & Ricardo Grieshaber-Bouyer[1,2,9,10,11] ✉

Neutrophils are frequently studied in mouse models, but the extent to which findings translate to humans remains poorly defined. In an integrative analysis of 11 mouse and 13 human datasets, we find a strong correlation of neutrophil gene expression across species. In inflammation, neutrophils display substantial transcriptional diversity but share a core inflammation program. This program includes genes encoding IL-1 family members, CD14, IL-4R, CD69, and PD-L1. Chromatin accessibility of core inflammation genes increases in blood compared to bone marrow and further in tissue. Transcription factor enrichment analysis implicates members of the NF-κB family and AP-1 complex as important drivers, and HoxB8 neutrophils with *JunB* knockout show a reduced expression of core inflammation genes in resting and activated cells. In independent single-cell validation data, neutrophil activation by type I or type II interferon, G-CSF, and *E. coli* leads to upregulation in core inflammation genes. In COVID-19 patients, higher expression of core inflammation genes in neutrophils is associated with more severe disease. In vitro treatment with GM-CSF, LPS, and type II interferon induces surface protein upregulation of core inflammation members. Together, we demonstrate transcriptional conservation in neutrophils in homeostasis and identify a core inflammation program shared across heterogeneous inflammatory conditions.

Neutrophils mediate homeostatic and inflammatory processes and display substantial phenotypic and functional heterogeneity. While animal models fuel fundamental discoveries in immunology, differences between humans and mice can impair the translation of findings[1]. To maximize impact on human health, life sciences increasingly benefit from seamless transitions between the mouse and human system. However, due to structural and functional differences in genomes, it is often unclear which aspects reflect conserved biology. Therefore, integrative analyses of cellular systems across species are important for the success of translational research.

Structurally, the mouse and human genomes are closely related. They harbor ~16,000 protein-coding genes considered to be one-to-

one orthologs with high confidence[2]. However, structural orthology does not equal functional similarity since expression patterns of orthologous genes can deviate substantially across organs and development[3]. In leukocytes, expression of most orthologous genes and lineage-specific genes, in particular, is well-conserved between humans and mice[4]. Despite this overall similarity, different species can display substantial differences in ortholog expression between tissues[5]. For example, human neutrophils are highly abundant in defensins, yet their mouse orthologs are expressed in gut epithelial cells, not in neutrophils. Furthermore, neutrophils display high phenotypic and functional heterogeneity as a function of organ, maturation, and inflammatory condition[6–9], but whether a core inflammation

program consisting of genes that become induced across a range of inflammatory conditions exists is not known. It is thus unclear how similarities and differences between human and mouse transcriptomes should be interpreted, particularly in the context of different inflammatory conditions.

To address these gaps in knowledge, we perform an integrative analysis of resting and inflamed leukocytes from humans and mice and assess the degree of conservation of gene expression. We find that human and mouse transcriptomes can be analyzed together and that lineage-specific gene expression was closely related between humans and mice. We further study how the neutrophil transcriptome changes in inflammation, using a wide range of studies covering in vitro and in vivo inflammation as well as resting conditions in human[10–21] and mouse[12,22–31]. While transcriptional responses to different activating stimuli are heterogeneous, we identify a core inflammation program in neutrophils conserved across species and conditions. We predict upstream regulators and find increasing accessibility of core inflammation program members in ATAC-seq. *JunB*[−/−] HoxB8 cells display a lower upregulation of core inflammation genes when stimulated with zymosan compared to wild-type cells. In single-cell RNA-seq data from resting and activated neutrophils, stimulation with type I and II interferon, G-CSF, *E. coli* is associated with higher expression of core inflammation genes. Further, neutrophils from COVID-19 patients with more severe disease display higher expression of core inflammation genes. Finally, we validated members of the core inflammation program using flow cytometry of stimulated human and mouse neutrophils and identified an interplay between tissue of origin and stimulation in driving the phenotype of the neutrophil inflammatory response. Our approach illustrates that multiple datasets of mouse and human gene expression data can be effectively combined to identify patterns shared across conditions and conserved across species. This approach can be transferred to other cell types and organisms to facilitate studies comparing gene expression across species.

## Results

### Integrative analysis of leukocyte gene expression across species

To assess gene expression similarities and differences between human and mouse immune cells, we obtained bulk RNA-seq data from six sorted leukocyte lineages from the Haemopedia atlas[12,32] (Supplementary Fig. 1). This dataset consisted of a total of 76 samples of T cells, B cells, dendritic cells, monocytes, NK cells, and neutrophils (Supplementary Fig. 1). Sequencing depths for samples across all lineages are shown in Supplementary Fig. 2a, b, and detailed quality control metrics are summarized in Supplementary Data 1. We then integrated gene expression matrices by mapping protein-coding, one-to-one orthologous genes with high confidence, according to ENSEMBL[33].

To evaluate the robustness of this approach, we performed a principal component analysis on the integrated expression matrix. For each lineage, up to 200 lineage-associated genes were selected. Here, sample distribution was driven predominantly by lineage, followed by species (Fig. 1a). As envisioned, lineage-associated gene expression was highest in each respective lineage and occurred across species in all lineages (Fig. 1b). Similarly, clustering of sample-wise Pearson correlation coefficients based on these genes was driven predominantly by lineage, confirming that in our analytical approach, lineage identity dominates species differences (Fig. 1c).

Correspondingly, expression of key lineage-associated genes was highly conserved between humans and mice (Fig. 1d), such as *CSF3R* and *CHI3L1* in neutrophils, *CD19* and *CD22* in B cells, CD3 molecules and *CD28* in T cells, *NKG7* and *GZMA* in NK cells, *MSR1* and *SERPINB2* in monocytes and *FLT3* and *MYCL* in dendritic cells. The highest correlation between human and mouse gene expression was observed in neutrophils ($r = 0.79$), followed by T cells (0.65), B cells (0.65),

Monocytes (0.56), and a weaker correlation in NK cells (0.24) and dendritic cells (0.22) (Fig. 1d).

This analysis demonstrates that mapping one-to-one orthologs allows an integrated analysis of leukocyte transcriptomes across species to identify conserved and divergent expression patterns of structurally related genes. Of note, although these data indicate a higher correlation in neutrophils compared to other lineages, this effect may have been influenced by smaller library complexities in neutrophils.

### Transcriptional conservation in resting neutrophils

To systematically analyze which genes display similar and divergent expression across species, we integrated transcriptional profiles of resting (not activated) neutrophils available through the Sequence Read Archive (SRA). In this context, resting neutrophils were defined as those isolated from blood or tissue in the absence of disease or experimental manipulation. In a total of 84 human and 39 mouse samples, we observed a high correlation in overall gene expression, transcription factor expression, and lineage-associated gene expression across humans and mice (Pearson's $r$ between 0.78–0.87, $P < 2.2 \times 10^{-16}$) (Fig. 2a). These results were remarkably similar to those obtained from the more homogenous Haemopedia dataset, further illustrating the robustness of this approach even when integrating multiple datasets from different sources.

We next focused on neutrophil lineage-associated genes and defined five *GENE: Gene* (*HUMAN: Mouse*) pairs based on their expression patterns. In addition to one-to-one orthologs, we considered high-confidence one-to-many and many-to-many orthologs.

Orthologs with high expression in both humans and mice included the key neutrophil genes *CSF3R* (encoding the G-CSF receptor), *CXCR2*, *NCF4* (neutrophil cytosolic factor 4), the transcription factors *MCL1*, *SPI1* (encoding PU.1, an essential transcription factor for terminal granulopoiesis[34,35]) and *JUNB*, a transcription factor prominently expressed in late neutrotime which plays a vital role in the inflammatory response of neutrophils[9,36] (Fig. 2b). As *CSF3R*, *CXCR2* and *JUNB* expression changes along neutrophil development, their concordance in expression might suggest that the analyzed neutrophils from humans and mice were of comparable developmental stage.

Orthologs with higher expression in human neutrophils included *FCGR3A* and *FCGR3B* (encoding CD16A and CD16B, respectively), which both are one-to-many orthologs of mouse *Fcgr4*. This group also included the receptor for activated complement (*C5AR1*) and *CXCR1*, the receptor for CXCL8 (human)/KC (mouse). Genes with higher expression in mouse neutrophils included the protease *Mmp9*, *Camp* (encoding Cathelicidin Antimicrobial Peptide), *Il1b*, and *Retnlg* (encoding Resistin-like gamma) (Fig. 2b).

Of note, most genes in categories 1–3 were one-to-one orthologs, although 13/133 (9.8 %) were one-to-many orthologs. However, well-known neutrophil genes without one-to-one orthologs were also identified (categories 4 and 5) and included *CXCL8* in humans, a cytokine abundantly expressed in blood neutrophils, and *Ccl6*, one of the most abundant chemokines in mouse neutrophils (Fig. 2b). Enrichment for neutrophil-related GO terms was found across all five groups of genes (Supplementary Fig. 3).

Thus, while resting human and mouse neutrophils display conserved expression of many key neutrophil genes and transcription factors, gene expression can deviate substantially in the same lineage between species, even for structurally highly related genes.

### A core inflammation program is shared across conditions and conserved across species

We next assessed how the expression of one-to-one structural orthologs changes in different inflammatory contexts. Neutrophils display varied phenotypes in homeostasis and inflammation[6,7,9,37], but it is unknown if a proportion of the transcriptional characteristics of

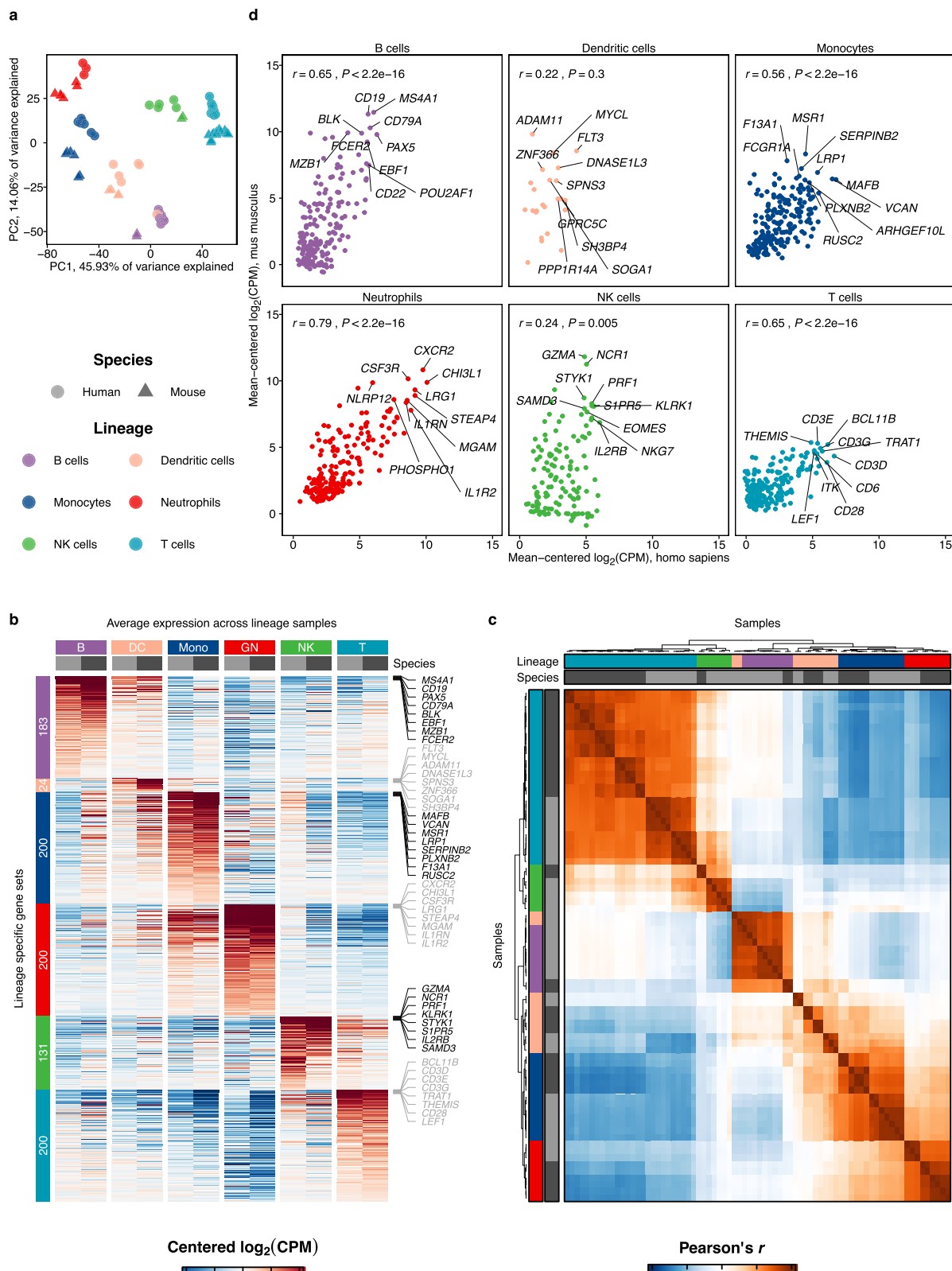

different neutrophil states is shared across different inflammatory conditions[9]. Here, resting neutrophils were defined as above and compared with their respective inflammatory condition.

To identify changes in inflammation, we analyzed 11 studies encompassing a total of 46 resting and 66 activated neutrophil samples across different conditions (Fig. 3a, Supplementary Data 2). We

tested for differential expression of genes with high-confidence one-to-one orthologs according to ENSEMBL separately within each study, comparing all reported conditions against their own resting controls to reduce the effect of technical variation between studies.

Compared to controls, inflamed neutrophils displayed 975 (median) differentially expressed genes (adjusted $P < 0.05$, absolute

**Fig. 1 | Integrative analysis of leukocyte gene expression across species.**
**a** Principal component analysis based on per-species mean-centered log$_2$(CPM) of lineage-specific genes shows a distribution driven predominantly by lineage.
**b** Concordant expression of the top lineage-specific genes for each lineage in each species. Shown is the average of log$_2$(CPM) centered expression values in each lineage. The number of lineage-specific genes shown is indicated on the left, up to 200 genes are shown per lineage. For each lineage, 8 genes with the highest expression in the respective cell line are labeled. **c** Lineage-specific gene expression dominates the species effect. Shown is the clustering of Pearson's $R$ based on the centered expression of top lineage-specific genes. **d** Neutrophils display the strongest correlation of lineage-specific gene expression across humans and mice compared to other leukocyte lineages. Gene expression (mean-centered log$_2$(CPM)) of lineage-specific genes was defined as above. Pearson correlation coefficient and $P$-value (two-sided) between human (x) and mouse (y) gene expression are shown on the top left (B cells: $r = 0.65$, $P < 2.2e-16$; Dendritic cells: $r = 0.22$, $P = 0.3$; Monocytes: $r = 0.56$, $P < 2.2e-16$; Neutrophils: $r = 0.79$, $P < 2.2e-16$; NK cells: $r = 0.24$, $P = 0.005$; T cells: $r = 0.65$, $P < 2.2e-16$). The top 10 most abundantly expressed genes are labeled. Source data are provided as a Source Data file.

log$_2$ fold change ≥ 0.5). These comprised 621 (median) significantly increased and 205 (median) significantly decreased genes (Supplementary Fig. 4a). Both the number of differentially expressed genes and the genes themselves were heterogeneous—concordant with the diverse transcriptional responses neutrophils can undergo in inflammation.

We next searched for potential overlap in the inflammatory response shared across conditions. Such an overlap may represent a "core inflammation program", from which neutrophils preferentially upregulate genes across a broad range of activating conditions.

We used Fisher's combined test to obtain a combined test statistic for each gene, summarizing individual comparisons from all datasets (Supplementary Data 3). Based on the elbow of the $P$-value-rank plot, we selected from the top 500 genes with the lowest $P$-value those with absolute log$_2$ fold change ≥0.5 (Fig. 3b).

A total of 221 genes displayed consistent changes in inflammation across studies: 179 genes were upregulated across comparisons (the "core inflammation program"), and 42 genes were downregulated (Fig. 3c). Effect sizes of those 221 up- and downregulated genes agreed well across all tested comparisons and across species (Fig. 3c, Supplementary Fig. 4b).

Core inflammation genes included the IL-1 molecules *IL1A* and *IL1B*, the LPS co-receptor *CD14*, the adhesion molecule *ICAM1*, the lectin receptor *CD69*, *CD40*, *IL4R* and *CD274* (encoding PD-L1) (Fig. 3c, d). Downregulated genes in inflammation included the cyclin-dependent kinase *CDK5R1*, *TLR5* (encoding Toll Like Receptor 5, an essential pathogen recognition receptor[38]), *CXCR4*, *CD101*, and the member of the mitogen-activated protein kinase family *MAP3K15* (Fig. 3c, d).

As expression of CD101 and CXCR4 changes throughout neutrophil maturation and aging, we compared the fold change of these markers between neutrophils activated in vitro and those activated in vivo to rule out the effects of differential release from the bone marrow under stress. No differences were observed in either marker (Supplementary Fig. 4c), suggesting that the transcriptional downregulation of CXCR4 and CD101 observed during neutrophil activation are cell-intrinsic and do not reflect a different maturation stage of neutrophils captured in the in vivo studies.

On the level of individual samples, we could confirm that the group of 179 core inflammation genes had either weak or absent expression in healthy neutrophils and were induced in inflamed neutrophils (Fig. 3d).

Gene set enrichment analysis identified a conserved enrichment of pathways related to apoptosis, inflammatory response, IL-2 and IL-6 signaling, IFN-γ response, and TNF signaling via NFKB and KRAS signaling (Fig. 3e).

Taken together, this integrative analysis of resting and activated neutrophils nominated a core inflammation program in neutrophils which is shared across inflammatory conditions and across species.

### The core inflammation program is detectable using different analytical strategies and in single-cell data

To further test the robustness of the core inflammation program, we performed two independent analyses. Using a linear mixed model, we observed high replicability of our results, with differentially expressed genes (absolute $\beta \geq 1$, $P_{adj} < 0.05$) identified by the linear mixed model

showing a strong skewing toward low Fisher $P$-values and a $\pi_1$-statistic of 0.71 (Supplementary Fig. 5).

We additionally assessed the replicability of differentially expressed genes between all tested comparisons. Median values of the $\pi_1$-statistic ranged from 0.06 to 0.60, depending on the study, and, importantly, did not show systematic species-driven differences (Supplementary Fig. 6a). Normalized enrichment scores for differentially expressed gene sets were in concordance with up-/down-regulation of the tested sets across all studies, supporting the existence of a shared core inflammation program. Of note, the downregulation of specific genes in inflammation was more variable across studies and hence less informative (Supplementary Fig. 6b). Pearson correlation coefficients of log$_2$ fold change values showed strong positive skewing, again pointing toward a core inflammatory response across conditions and species (Supplementary Fig. 6c).

As an additional analytical approach, we performed a weighted correlation network analysis (WGCNA)[39]. WGCNA constructs correlation networks and can help to identify clusters of genes ("modules") that are co-expressed across different conditions. It identified four modules (19, 5, 8, and 4) with significant enrichment for core inflammatory response genes (Fisher's exact test, $P_{adj} < 0.05$). Gene expression within those four modules increased in inflammation and contained several members of the core inflammation program (Supplementary Fig. 7).

For validation purposes, we analyzed four recent single-cell RNA-sequencing datasets that had not been used to derive the core inflammation program. These included neutrophils from healthy control individuals and those with mild to moderate or severe COVID-19 (Combes et al., dataset 1)[40], human neutrophils stimulated with G-CSF, IFN-β or IFN-γ (Montaldo et al., datasets 2+3)[41] and mouse neutrophils infected with *E. coli* (Xie et al., dataset 4)[7].

Expression of most of the 179 core inflammation genes increased in inflamed neutrophils (Fig. 4a). A gene set was created based on the 179 core inflammation genes, and changes in expression were tested compared to random background genes with the same expression abundance. A significant increase in the core inflammation genes was detected in all conditions and was higher in patients with severe compared to mild to moderate COVID-19 (Fig. 4b). However, examination of the expression of the core inflammation program on a single cell level indicated heterogeneity within the population of neutrophils, which was characterized by the presence of groups of cells with exceptionally high or low expression of the defined gene set in inflamed states (Fig. 4c).

### The core inflammation program shows conserved transcriptional regulation across species

To identify putative regulators of neutrophil activation in inflammation, we applied transcription factor (TF) enrichment analysis individually to up- and downregulated genes in each study. TF enrichment across mouse and human inflamed neutrophils was highly consistent in TFs with decreasing (Supplementary Fig. 8a) and increasing (Supplementary Fig. 8b) activity.

Transcription factors that we found to be enriched in genes expressed in resting neutrophils include AKNA, PU.1 (encoded by *SPI1*), FOXO3, FOXO1, TFEB, RARA, and STAT5B (Supplementary Fig. 8a). Transcription factors that we found to be enriched in genes associated

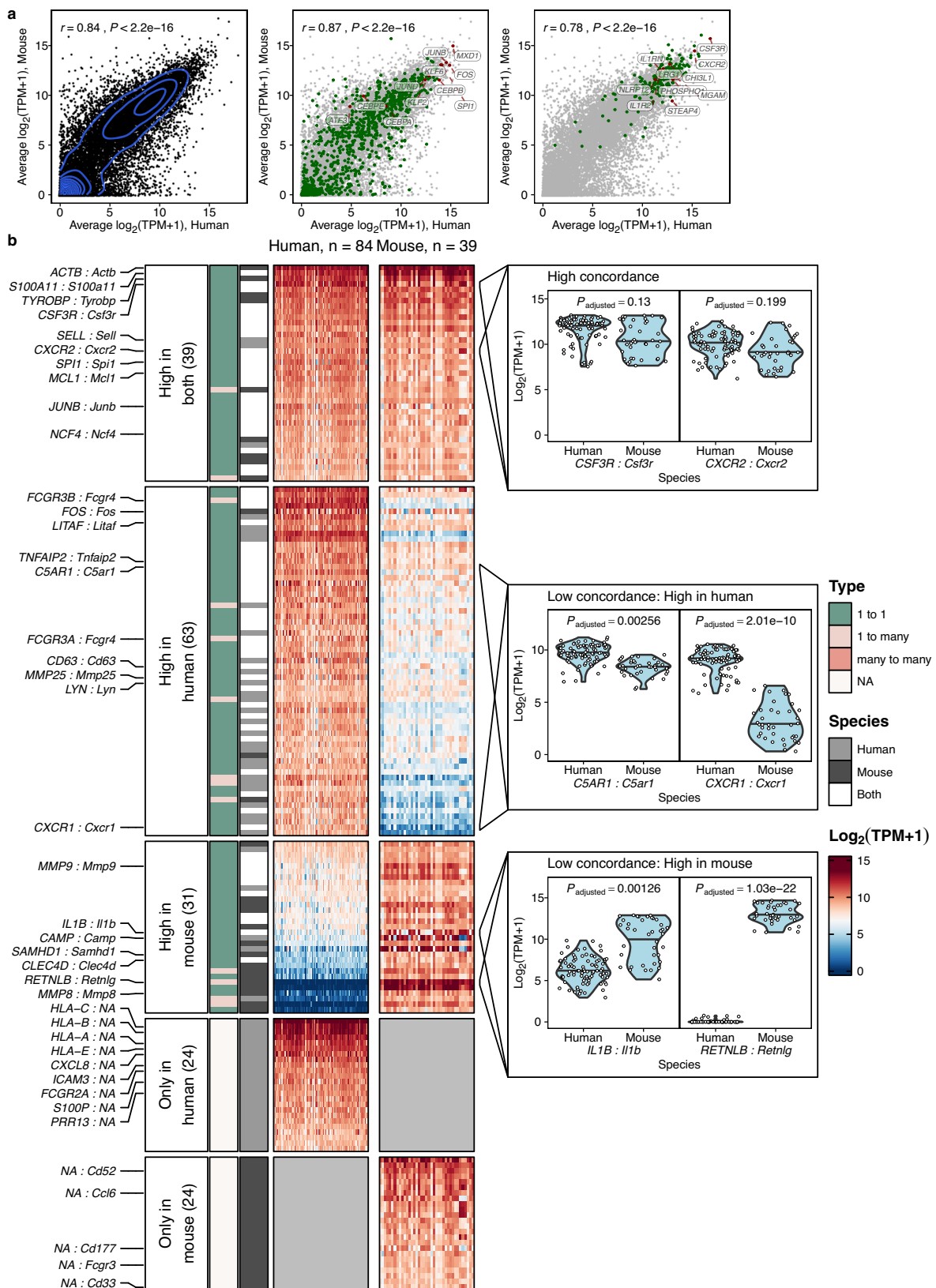

with inflamed neutrophils included CSRNP1, PLSCR1, FOS, FOSB, the NF-κB components NFKB1/NFKB2, the emergency granulopoiesis transcription factor CEBPB and JUNB (Supplementary Fig. 8b).

To reduce this selection of transcription factors to those with the highest changes in inflammation, we compared the predicted regulatory activity of transcription factors and their respective gene expression in inflammation. This analysis highlighted that the genes encoding for CSRNP1, JUNB, CEBPB, XBP1, and ETS2 were strongly upregulated in inflamed neutrophils while also displaying strongly increased regulatory activity (Fig. 5a).

On the level of individual studies, we also found high consistency in the transcription factors predicted to be enriched in genes

**Fig. 2 | Conservation of neutrophil gene expression in homeostasis. a** Strong correlation of gene expression between resting human (x) and mouse (y) blood neutrophils. Left: all genes ($r = 0.84$, $P < 2.2e\text{-}16$). Middle: transcription factors, highlighted in green ($r = 0.87$, $P < 2.2e\text{-}16$). Transcription factors were retrieved from a curated set of transcription factors in ChEA3. The top 5 TFs (based on the sum of the average expression in human and mouse) were labeled and highlighted in red. Additionally, we manually labeled and highlighted the genes *JUND*, *KLF2*, *ATF3*, *CEBPA*, *CEBPB*, and *CEBPE*. Right: lineage-specific genes as depicted and defined in Fig. 1, highlighted in green ($r = 0.78$, $P < 2.2e\text{-}16$). Neutrophil genes were labeled as in Fig. 1c and highlighted in red. Shown are ($\log_2(\text{TPM}+1)$) expression values. Pearson correlation coefficients between human and mouse gene expression for the three groups as well as *P*-values (two-sided) are shown in the upper left of each panel. **b** Neutrophil lineage-associated genes with orthologs can show

concordant or discordant expression across species. Gene expression heatmap ($\log_2(\text{TPM}+1)$) of neutrophil lineage-associated genes that were assigned to five different expression profile groups: high expression in both species, high expression in human/mouse and low in the other species, high in human/mouse and no high-confidence ortholog; see "Methods". Gene-gene pairs of particular importance in neutrophils are highlighted (HUMAN SYMBOL: Mouse Symbol). Annotated are Orthology relationships between the respective genes as well as species in which the gene was detected as lineage-associated. Right, violin plots of selected gene-gene-pairs show their expression in individual samples for each species. Benjamini-Hochberg adjusted *P*-values derived from a gene-wise likelihood ratio test between two linear mixed models with and without species as fixed effect are shown for each highlighted gene-gene pair; see "Methods". Source data are provided as a Source Data file.

---

upregulated and downregulated in activated neutrophils (Supplementary Fig. 8c). These results were consistent with an independent enrichment analysis performed separately for each species (Supplementary Fig. 8d, e).

## Migration into tissue and activation significantly enhance chromatin accessibility and expression of core inflammation genes

If genes in the core inflammation program are predisposed to be upregulated, then chromatin accessibility for these genes should increase upon neutrophil maturation, migration into tissues, and exposure to inflammatory stimuli.

To test this hypothesis, we analyzed chromatin accessibility data derived from bone marrow, blood, and an air pouch model of acute inflammation. These data were generated using Assay for Transposase-Accessible Chromatin using sequencing (ATAC-Seq), a method that tests genome-wide chromatin accessibility. Briefly, ATAC-seq allows the analysis of chromatin accessibility by sequencing DNA fragments that are bound by a hyperactive Tn5 transposase, which preferentially inserts sequencing adapters into open chromatin regions[42]. In the air pouch model (executed on C57BL/6J mice), blood neutrophils first migrate into a sterile membrane in the skin before being activated by zymosan in the air pouch[36]. Of the 179 core inflammation program genes, 29 displayed increasing accessibility in blood vs. bone marrow, compared to only 10 genes with decreased accessibility (Fig. 5b). Neutrophils that had transmigrated from the blood into the membrane displayed enhanced accessibility of 78 genes. This increase was significantly ($P = 5.1 \times 10^{-9}$) higher than the increase of 29 genes for neutrophils in blood compared to bone marrow.

Similar skewing toward enhanced accessibility of core inflammation genes was observed when membrane vs. bone marrow (89 up, $P = 1.2 \times 10^{-13}$), inflamed air pouch vs. blood (85 up, $P = 1.5 \times 10^{-11}$), and inflamed air pouch vs. bone marrow (100 up, $P = 2.1 \times 10^{-17}$) were compared to blood vs. bone marrow. (Fig. 5b). The number of genes with increased accessibility was significantly higher than expected by chance, as compared to the accessibility of randomly selected background genes. Importantly, the genes with increased and decreased accessibility were highly consistent across the comparisons (Fig. 5c).

After finding that core inflammation genes have increased chromatin accessibility even before the onset of inflammation, we searched for potential driver transcription factors displaying increasing expression and regulatory activity in inflammation. Comparing motif enrichment (HOMER) with actual expression change in air pouch vs. blood, we observed an increase in both measures for a remarkably restricted set of transcription factors, namely ATF3, BATF, FOSL1, JUNB, and JUN (Fig. 5d).

We next investigated whether the core inflammation program represents a group of genes from which neutrophils preferentially draw upon exposure to inflammatory stimuli. If this were the case, then it would be more likely for core inflammation genes to be

upregulated in inflammation compared to all other genes. We analyzed RNA-seq data from differentiated HoxB8 neutrophils stimulated with or without zymosan for 2 h[36]. We observed that in activated neutrophils, a significantly higher proportion of core inflammation genes ($107/179 \approx 60\%$) was upregulated than expected by chance (36–74 genes in 1000 simulations using expression-matched background genes; $P_{overrepresentation} = 6.5 \times 10^{-41}$) (Fig. 5e).

When evaluating predicted conserved regulatory activity and change in chromatin accessibility together, JUNB emerged as a prominently affected transcription factor and has previously been shown to control neutrophil activation[36] and to be highly expressed upon neutrophil activation[43]. On the other hand, CEBPB has previously been shown to be a key transcription factor mediating emergency granulopoiesis[44] and showed a high predicted regulatory activity in our analysis with limited changes in chromatin accessibility. To assess the impact of two transcription factors identified in our enrichment analysis on the expression of core inflammation program genes, we repeated the same analysis in differentiated HoxB8 neutrophils carrying a genetic knockout of either *JunB* or *Cebpβ*. CEBPB showed upregulation in inflamed neutrophils as well as increased regulatory activity. In addition, JUNB, which plays an important role in the inflammatory response of neutrophils[9,36], also had increased motif enrichment in the air pouch vs. blood comparison.

Based on these analyses, we expected a modest reduction in the expression of core inflammation genes in *Cebpβ*$^{-/-}$ cells and a stronger reduction in *JunB*$^{-/-}$ cells. Indeed, this was the case: In a direct comparison of resting knockout (*JunB*$^{-/-}$ and *Cebpβ*$^{-/-}$) versus wild-type cells, we observed a significantly stronger downregulation of the core inflammation program in *JunB*$^{-/-}$ cells (69 genes; $P = 1.5 \times 10^{-9}$) than in *Cebpβ*$^{-/-}$ cells (43 genes; $P = 0.0011$) (Fig. 5e and Supplementary Fig. 9).

Comparing zymosan-stimulated knockout cells versus wild-type cells, we again saw a significant downregulation of core inflammation genes in the *JunB*$^{-/-}$ condition (51 genes; $P = 0.0025$) but not in the *Cebpβ*$^{-/-}$ condition (25 genes; $P = 0.79$) (Fig. 5e and Supplementary Fig. 9).

Together, these results indicate that maturation and migration into an inflamed tissue site predispose neutrophils to upregulate genes of the core inflammation program and that knockout of *Cebpβ* and especially *JunB* leads to a weaker induction of core inflammation genes compared to WT cells.

## Members of the core inflammation program can be validated on the protein level in activated human and mouse neutrophils

To validate members of the core inflammation program experimentally, we filtered the list of genes by surface proteins, yielding 36 markers (Fig. 6a)[45]. Based on antibody availability, we developed a flow cytometry panel including canonical lineage markers (human: CD15, mouse: Ly6G) and five proteins predicted to be part of the core inflammation program: CD14, CD69, CD40, CD274 (PD-L1) and IL-4R (Supplementary Tables 1 and 2).

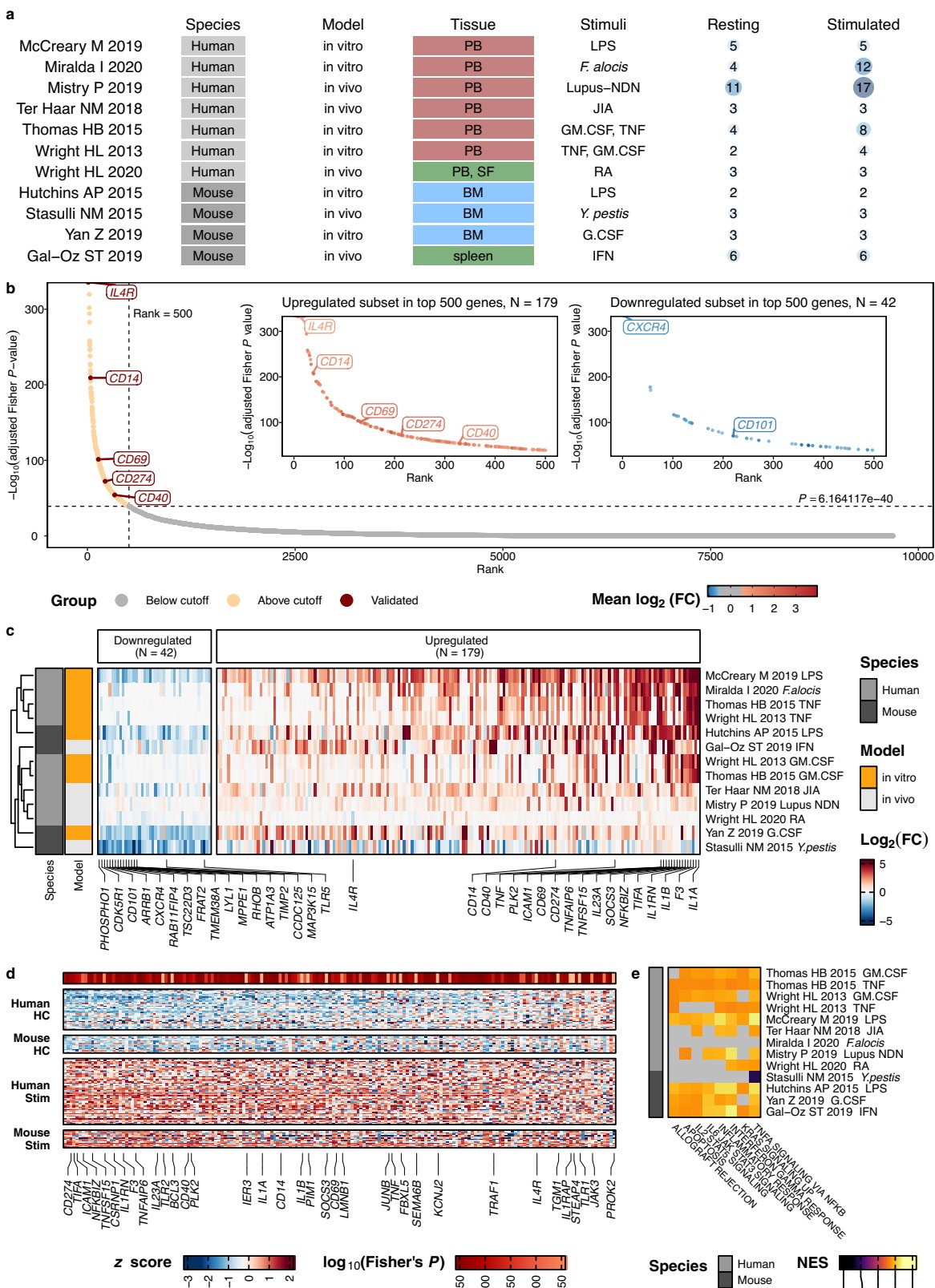

We isolated human neutrophils from peripheral blood and mouse neutrophils from bone marrow and cultured them over 48 h with or without the addition of GM-CSF + LPS and GM-CSF + IFN-γ (Fig. 6b).

Prolonged cell culture without activation led to an increase in CXCR4 and loss of CD62L and CD101 in human cells, while mouse cells showed a reversed phenotype with upregulation of CD62L and CD101

as well as a downregulation of CXCR4, suggesting continued maturation of bone marrow neutrophils in vitro and not classical neutrophil aging (Supplementary Fig. 10). Compared to unstimulated cells, activated mouse neutrophils significantly upregulated the predicted core inflammation program markers CD14, CD40, CD69, PD-L1, and IL-4R in the condition containing LPS and all but CD69 in the condition

**Fig. 3 | A core inflammation program is conserved across mouse and human neutrophils. a** Overview of the 11 studies integrated for analysis. Differential expression testing was performed independently for each study, resting neutrophils within each study were used as control. **b** A combined analysis of the neutrophil response to activation/inflammation identifies 179 consistently upregulated (core inflammation program) and 42 downregulated genes in inflammation. Shown are all ($N = 9697$) tested genes ranked by their $-\log_{10}$ Benjamini-Hochberg adjusted Fisher $P$-value (adjusted Fisher's combined test on two-sided $P$-values from individual differential expression analyses for each comparison). The 500 genes with the lowest $P$-values were subjected to an additional filtering step based on a $\log_2$ fold change cutoff $\geq 0.5$ and $\leq -0.5$ for upregulated and downregulated genes. Highlighted (*IL4R*, *CD14*, *CD69*, *CD274*, *CD40*) upregulated genes were validated experimentally (Figs. 6 and 7). **c** 42 genes downregulated in inflammation and 179 core inflammation genes are shared across studies. Shown are the $\log_2$ fold changes across comparisons of genes up- and downregulated in inflammation. Rows represent a comparison, columns represent genes that passed our meta-analysis thresholds. Columns are arranged by the mean $\log_2$ fold change across all comparisons. For each direction, the 15 genes with the highest absolute $\log_2$ fold change are labeled, as well as genes encoding for proteins validated in Figs. 6 and 7. **d** Core inflammation genes are not expressed in resting neutrophils in both species and are induced upon activation. Shown is a heatmap with relative expression values ($z$-score for each gene across samples) of the core inflammation genes. Each column represents a gene, and each row a sample. $P$-values: Results of a Benjamini-Hochberg adjusted Fisher's combined test. We labeled the top 20 genes with the lowest $P$-values, genes that were also labeled in (**c**), and manually labeled *TRAF1* and *JUNB*. **e** Conserved Gene Set Enrichment Analysis based on rankings derived from each comparison's $\log_2$ fold changes. Heatmap showing normalized enrichment scores. Only pathways that have been significant in more than 50% of comparisons are depicted. Gray fields indicate nonsignificant NES values. Source data are provided as a Source Data file.

containing IFN-γ (Fig. 6c). Human neutrophils displayed a highly concordant increase in those markers. CD69 and PD-L1 increased with similar magnitude, while upregulation of CD14 was stronger in mouse neutrophils compared to human neutrophils. In human neutrophils, upregulation of CD40 was restricted to a small (~2%) population of neutrophils (in line with previous findings[46]) but reached significance on the bulk level for both stimulations (Fig. 6c).

Differences were also noticeable between inflammatory conditions. In mouse neutrophils, the combination of GM-CSF and LPS led to a stronger increase in the expression of CD14, CD69, IL-4R, and CD40 compared to GM-CSF and IFN-γ. The reverse was true for PD-L1, which is driven substantially by IFN-γ signaling[8]. Further, IFN-γ stimulation reduced CD69 expression, while LPS increased it. In human neutrophils, the combination of GM-CSF and IFN-γ leads to stronger increases in CD14, CD69, IL-4R, and PD-L1 than the combination of GM-CSF and LPS.

A combined diffusion map analysis revealed a high degree of overlap between mouse and human neutrophils, while cell distribution was driven predominantly by experimental conditions (Fig. 6d). Correspondingly, activated neutrophils of both species displayed a continuous upregulation of the inflammatory response markers (Fig. 6e).

These findings confirm the predicted activation markers, further substantiating the conservation of inflammatory response programs in neutrophils while also revealing differences between species and inflammatory conditions.

**Neutrophil origin and inflammatory condition influence the expression of the core inflammation program**

Neutrophil heterogeneity is influenced by the tissue microenvironment[6,9]. To evaluate the impact of tissue origin on the phenotype of neutrophils in inflammation, we performed stimulation experiments with paired leukocyte preparations from blood, bone marrow, and spleen of wild-type BL6 mice. In a principal component analysis of flow cytometry data, resting neutrophils clustered closely together, but each tissue remained distinguishable based on subtle baseline expression differences in IL-4R, CD69, and CD40 (Fig. 7a). Inflamed neutrophils deviated markedly from their resting counterparts and reached distinct states as a function of tissue and inflammatory condition (Fig. 7a).

Neutrophils from all tissues upregulated CD69 and IL-4R, suggesting that these markers can be utilized as neutrophil activation markers across a variety of conditions (Fig. 7a). In contrast, expression of CD40, CD14, and PD-L1 showed greater tissue dependence. CD40 (evident most prominently in mouse neutrophils) was robustly upregulated in splenic neutrophils and less prominently in blood neutrophils. Conversely, CD14 and PD-L1 expression was inducible to a greater extent in blood neutrophils and bone marrow neutrophils but less in splenic neutrophils.

We also noted differences related to activating stimuli, for example, through more prominent PD-L1 induction by IFN-γ compared to LPS. The single-cell analysis highlighted a continuum of states in all organs (Fig. 7b), driven by increasing expression of the core inflammation markers (Fig. 7c). Importantly, the core program was already inducible in bone marrow neutrophils, suggesting that in vitro and adoptive transfer experiments performed with bone marrow neutrophils can recapitulate important features of neutrophil biology in inflammation.

## Discussion

Neutrophils are important mediators of immune defense and protagonists in immune-mediated diseases. Mouse and human neutrophils differ in morphology, frequency in blood (humans ~50–70%, mice ~10–25%), and expression of marker proteins. For example, mouse neutrophils are defined by surface expression of Ly6G, not present in the human genome, whereas mouse neutrophils lack expression of defensins[47].

Both in humans and mice, neutrophils are phenotypically heterogeneous across different tissues and inflammatory conditions[37,48,49]. Recent studies suggest that neutrophil heterogeneity in homeostasis is driven by a chronological sequence of maturation and activation termed neutrotime, whereas the combination of aging, tissue factors, environmental features, and inflammatory signals promote their polarization toward distinct states[6,7,9].

While the neutrotime signature can be detected in both species and this overarching principle of neutrophil ontogeny is likely conserved across humans and mice, it is poorly understood which features of the neutrophil inflammatory response are shared across species. Furthermore, it is unclear which aspects of the neutrophil inflammatory response reflect a general inflammatory response program shared across multiple inflammatory conditions and which features are highly specific to certain triggers or sites of inflammation.

To address these gaps in knowledge, we performed an integrative analysis of resting and inflamed RNA-seq samples from humans and mice. We validated our computational approach by comparing gene expression conservation across six immune cell lineages: T cells, B cells, monocytes, dendritic cells, NK cells, and neutrophils. Expression of lineage-specific genes was generally well-conserved across humans and mice. Intriguingly, neutrophils displayed both the greatest number of lineage-specific genes and the highest correlation of gene expression between mice and humans, suggesting a higher degree of conservation in this phagocytic cell compared to other lineages.

While different inflammatory conditions induced highly heterogeneous responses in neutrophils, our combined analysis allowed us to predict a core inflammation program conserved across mice and humans. The robustness of this program was underscored by the high concordance between the gene set derived from Fisher's combined test and complementary approaches based on a linear mixed model as

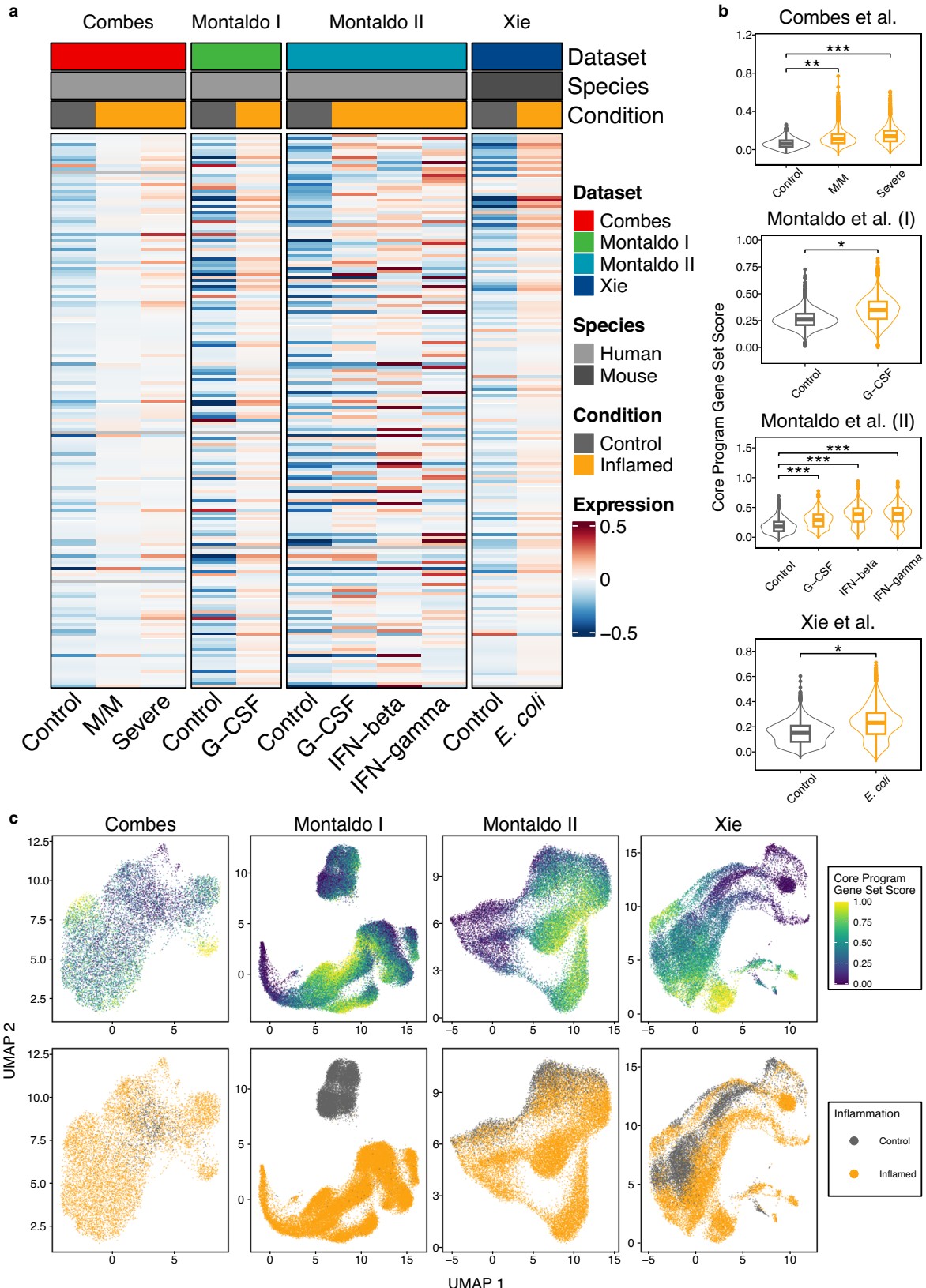

well as weighted correlation network analysis (WGCNA[39,50,51]). It is important to note that different analytical strategies may be used to derive this core inflammation program, each detecting a varying number of genes. The situation is similar for differential gene expression in general, which depends on the chosen method, as has been reviewed extensively[52]. Nevertheless, our analysis indicates that a group of genes exists from which neutrophils preferentially draw when they become activated across humans and mice and across a large range of conditions and disease states. The conservation of a small set of transcription factors predicted to regulate a broad variety of conditions across humans and mice highlights the conserved nature of gene expression in neutrophils.

**Fig. 4 | Validation of the enrichment of core inflammation genes in single-cell RNA sequencing.** The first dataset (Combes) includes neutrophils of patients without (control), mild (M/M), and severe cases (severe) of COVID-19. The second dataset (Montaldo I) was derived from in vivo neutrophils from healthy controls (control) and patients treated with G-CSF (G-CSF). The third dataset (Montaldo II) used in vitro stimulation of umbilical cord blood neutrophils with G-CSF, interferon-β, or interferon-γ. The fourth dataset (Xie) consisted of neutrophils isolated from different organs of mice challenged with *E. coli*. **a** Core inflammation genes show a higher expression in inflamed samples compared to controls. Shown is a heatmap with mean scaled expression values of the core inflammation genes across all cells per condition and sample. Rows represent genes, columns samples. **b** Inflamed cells show a higher gene module score for the core inflammation program. Each cell has been scored for the enrichment of the core inflammation program genes compared to a random reference gene set of similar expression. Shown are control samples in gray alongside inflamed cells in orange. *P*-values were derived from a maximum likelihood ratio test of linear mixed models. Box plots: Median between the 25th and 75th percentile, whiskers extend to 10% and 90%. Outliers are shown as dots. Top plot: Statistics are derived from a total of 10,782 cells (Control: 1026, M/M: 5732, Severe: 4024) cells from $N = 56$ independent samples ($P_{\text{Control:M/M}} = 0.0109$, $P_{\text{Control:Severe}} = 0.000013$). Top mid: The experiment included a total of 48,875 cells (healthy: 12,338, G-CSF: 36,537) from $N = 6$ independent samples ($P_{\text{Control:G-CSF}} = 0.01416$). Bottom mid: A total of 26,312 cells (Control: 4296, G-CSF: 5049, IFN-beta: 8472, IFN-gamma: 8495) from $N = 4$ independent samples were analyzed ($P_{\text{Control:G-CSF}} = 9.42 \times 10^{-5}$, $P_{\text{Control:IFN-beta}} = 2.67 \times 10^{-5}$, $P_{\text{Control:IFN-gamma}} = 2.71 \times 10^{-5}$) Bottom plot: Calculation was performed on 26,239 cells (Control: 8990, *E. coli*: 17,249) from $N = 10$ independent samples ($P_{\text{Control:}E.\ coli} = 0.0376$). **c** UMAP embedding. Coloring by gene set score (top panel) and the cell's inflammatory state (bottom panel) shows an increase in the core inflammation program expression in inflamed samples compared to their control. Source data are provided as a Source Data file.

To validate the predicted core inflammation program in different models, we analyzed differential gene accessibility in ATAC-sequencing data from a mouse air pouch model of inflammation. We found a significant proportion of core inflammation program genes to be more accessible with maturation and in pro-inflammatory conditions. While this model is very specific, it covered neutrophils from different maturation stages and presented the opportunity to study transmigrated and activated neutrophils separately. Further, analysis of the transcriptome on a single cell level in both in vivo and in vitro inflamed neutrophils of both species allowed us to validate the core inflammation program. While the overall enrichment of the proposed gene set on a pseudo-bulk level was clearly evident, our analyses also suggested significant heterogeneity within the population of inflamed neutrophils, consistent with recent analyses[7,9,40,41]. These analyses further highlight the predictive value of the program in a method not used in its generation.

HoxB8-derived neutrophils are a powerful tool to model neutrophil function. We assessed the differential expression of zymosan-activated HoxB8-derived neutrophils versus control, showing a significant overrepresentation of core inflammation genes in activated neutrophils. Zymosan-activated myeloid cells through TLR2 and is a commonly used pro-inflammatory trigger. The core inflammation program was reduced in resting cells carrying a knockout of key regulators of this program (*JunB*$^{-/-}$ and *Cebpβ*$^{-/-}$). Core inflammation genes were also significantly less upregulated in zymosan-stimulated *JunB*$^{-/-}$ cells, indicating an impaired neutrophil inflammatory response in these cell lines. Concordant with previous reports of a more limited impact of the *Cebpβ* knockout on inflammatory neutrophil functions compared to the *JunB* knockout[36], the underrepresentation of core inflammation genes was nonsignificant in our analysis.

Finally, we validated key components of the predicted core inflammation program experimentally. Using primary human and mouse neutrophils, we showed that the surface proteins CD14, CD69, IL-4R, CD40, and PD-L1 are induced by in vitro cytokine stimulation, and this upregulation is observable in both species, although CD40 was restricted to a small subset of neutrophils in humans, as expected[46].

This finding further underlines the conserved character of the inflammation program as presented in this study. Interestingly, while neutrophils from different mouse tissues upregulated the inflammatory response markers, the magnitude of upregulation differed across bone marrow, spleen, and blood, suggesting that the tissue origin of neutrophils is an important consideration in experimental studies.

Recently, Jin et al. identified a distinct neutrophil population termed "antigen-presenting aged neutrophils (APANs)"[53]. In humans, this population was characterized as CD66b$^+$CXCR4$^+$CD62L$^{lo}$CD40$^+$ CD86$^+$, while in mice, they were identified as Ly6G$^+$CXCR4$^+$CD62L$^{-/lo}$ MHCII$^+$CD40$^+$CD86$^+$. APANs were capable of inducing CD4 T cell proliferation via IL-12 and exhibited a hyper-NETosis phenotype. The presence of these neutrophils in patients with sepsis was associated with increased mortality. While we also observed the upregulation of key marker genes like *CD40* in our study's core inflammation program, APANs displayed distinct features, such as elevated levels of *CXCR4* and coexpression with *CD74*, suggesting a unique neutrophil polarization state discriminable from both neutrophil aging and canonical activation. The phenotype observed by the authors suggests the importance of further studying APANs, their features and their role in antigen presentation in humans and mice.

The upregulation of IL-4R we observed is concordant with reports of IL-4R upregulation during sterile information in mice, with implications for diseases that are IL-4 mediated[54]. CD14 has recently been shown to be an important, highly cell-specific mediator of TNF response in a mouse sepsis model[55]. Interestingly, CD14$^+$ macrophages and neutrophils were found to be key players leading to lethality in response to TNF (with improved survival in CD14-deficient mice), which provides a model for the cytokine storm seen in severe sepsis and provides evidence for the complexity of CD14-mediated inflammatory response beyond TLR-signaling. These examples highlight the importance of core inflammation program members and stress the need to study them in a broad variety of inflammatory contexts.

The question of how well mouse models mimic human immunology is an area of ongoing debate. Even the same data can support different conclusions[56,57], highlighting the impact of analytical decisions. Furthermore, it is important to compare suitable datasets, control for batch effects, and make comparisons to varying controls to avoid a shared denominator effect[56,58,59].

In the context of neutrophils, fundamental differences between humans and mice exist[60,61]. Those differences must be considered when using the mouse as a model to study neutrophil function, especially in disease, as previously discussed[62]. Granule proteins found in neutrophils play a key role in defense against infection. An important difference in the granule protein repertoire includes α-defensins, which exercise antimicrobe[63,64] and chemotactic[65] activity and are absent in mouse neutrophils. It is also known that mouse neutrophils express less MPO, leading to a more limited capability to produce hypochlorous acid compared to their human counterpart[66]. The importance of cytokine production by neutrophils has been increasingly recognized[67,68], with some cytokines such as IFN-β and IL-17 apparently expressed in mouse and not human neutrophils. The different immunoreceptor reservoir[69] is, in part, a result of pathogen responses that are exclusive to the human species. For example, human neutrophils express specific CEACAMs that mediate uptake of the human-specific pathogen *Neisseria gonorrhea*[70], which must be taken into account when modeling neutrophil responses to this pathogen[71]. Taken together, these studies provide important context to be taken into account when interpreting the core inflammation program identified.

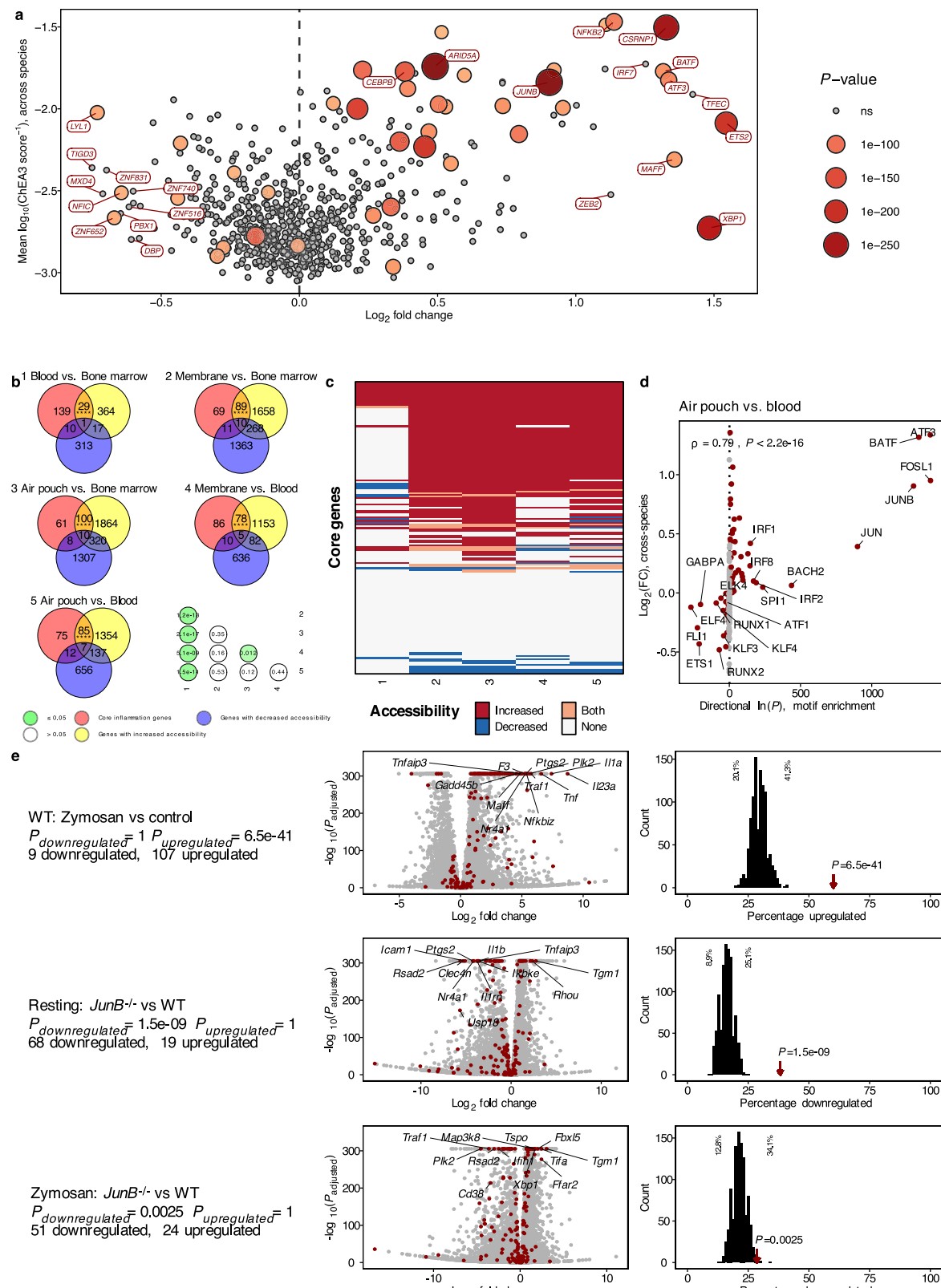

The derivation of the core inflammation program was limited to bulk RNA-sequencing samples since a similar analysis using single-cell studies requires datasets that are only now beginning to emerge. To circumvent potential batch effects, we focused our analysis on studies with internal controls of resting neutrophils, excluding other potentially interesting studies containing only neutrophils harvested from inflamed sites. Analyzed samples were also limited by technical factors, including the known intron retention in neutrophils[72], as well as the less complex transcriptome associated with low RNA and high RNase content. Furthermore, analysis of single-cell RNA sequencing data, the ATAC-seq data from the air pouch model of inflammation, and RNA-seq data from zymosan-activated HoxB8 samples represent only selected validation strategies in specific modalities of inflammation, which might limit the generalizability of some of the findings.

**Fig. 5 | Genes in the core inflammation program are predisposed to be upregulated with maturation and activation. a** Shown is the mean $\log_2$ fold change of gene expressions across all comparisons versus the regulatory activity (inverse logarithm of ChEA3 Scores per species). Colors and sizes indicate $-\log_{10}(P)$, using the Benjamini-Hochberg adjusted $P$-values from a Fisher's combined test (Fig. 3). **b–d** ATAC-sequencing from the air pouch model[36]. **b** Members of the core inflammation program show increasing chromatin accessibility (blood vs. bone marrow; membrane and air pouch vs. bone marrow or blood). Genes part of the inflammatory response program were compared with the list of genes that showed increased/decreased accessibility in the depicted comparisons. Asterisks indicate significance versus 1000 repeats of an accessibility analysis of random RNA-expression-matched background genes (two-sided studentized bootstrap: $P_1$: $7.26 \times 10^{-8}$; $P_2$: $6.86 \times 10^{-17}$; $P_3$: $2.09 \times 10^{-21}$; $P_4$: $1.56 \times 10^{-18}$; $P_5$: $3.07 \times 10^{-22}$). Bottom-right: Two-sided Fisher's exact test on the number of core inflammatory genes with increased accessibility versus no increase. FDR-adjusted $P$-values. **c** Core inflammation program gene accessibility for each comparison. Rows ordered by nested decreasing rank of peaks associated with an increase, with both, with none, and

with a decrease. **d** A subset of transcription factors shows increased motif enrichment and increased gene expression in inflamed neutrophils. Motif enrichment analysis was performed using HOMER and is compared to the mean $\log_2$ fold change across species. A one-sided motif enrichment analysis was run separately for increased and decreased accessibility to calculate $ln(P)$. $\rho$, Spearman's rank correlation coefficient with its respective $P$-value (two-sided). **e** Analysis of HoxB8 cells that were treated with zymosan[36]. Core inflammation genes show upregulation versus random genes in zymosan-stimulated HoxB8 cells and are downregulated in JunB$^{-/-}$. Core inflammation genes in HoxB8 neutrophil RNA-Seq data[36]. One row per comparison. Left, experimental conditions for each comparison and respective statistics. Middle, volcano plots for each comparison. Red, Core inflammation program members. Members with the highest combined significance and effect sizes are labeled. Right, histograms showing the percentage of expression-matched background genes (equally sized gene sets, 1000 simulations) up/downregulated (see $x$-axes) in each comparison. Red arrow indicates observed percentage for core inflammation program members, annotated with its $P_{overrepresentation}$ (one-sided, Wallenius method[99]). Source data are provided as a Source Data file.

Nevertheless, our combined analysis of 11 human and mouse neutrophil transcriptomic datasets identified a largely conserved transcriptomic landscape across species, supporting the use of mouse neutrophils to illuminate human biology. We furthermore predicted and experimentally confirmed the existence of a core inflammation program conserved across human and mouse inflamed neutrophils. This study sets the stage for more fine-grained analyses of the epigenome, transcriptome, and proteome of neutrophils across varying conditions, which together will paint a clearer picture of the neutrophil response to different environments. Going forward, genetic perturbations and pharmacological interventions to interfere with pathologic neutrophil activation will be particularly informative if focused on programs conserved across species. The systems biology approach presented here can be transferred to other cell types and organisms to facilitate further studies comparing gene expression across species.

## Methods

### Ethics approval

Research with healthy human participants followed the declaration of Helsinki. Blood of healthy donors was collected under IRB-approved protocols (Heidelberg S-272/2021 and Heidelberg S-285-2015) approved by the ethics committee of the University of Heidelberg, Heidelberg, Germany. Informed consent was obtained from all participants.

Experiments involving animals were conducted under the approval of the Animal Care Facility Heidelberg and the Animal welfare officers (approval #T66/21) at the University of Heidelberg, Heidelberg, Germany.

We obtained publicly available RNA sequencing data from mouse and human leukocytes through GEO (262 samples from 24 studies) and integrated these data by mapping orthologous genes. Differential expression analysis between resting and inflamed neutrophils was performed separately for each dataset, and the core inflammation program was derived using Fisher's combined test. Transcription factor enrichment analysis was performed using ChEA3 and DoRothEA and compared to chromatin accessibility data from ATAC-seq (GSE161765). The impact of *Cebpβ* and *JunB* knockout on the core inflammation program was studied using RNA-seq data from HoxB8 cells (GSE161765). The core inflammation program was validated in stimulated mouse and human neutrophils by flow cytometry.

### Datasets

For all analyses, we used the following datasets:

### RNA sequencing

Datasets of interest were identified through a literature search on PubMed and the NCBI Gene Expression Omnibus. In total, 262 publicly available RNA sequencing samples from 24 studies were included.

- Lineage atlas dataset (Table 1): 76 samples, 40 human samples, 36 mouse samples. This dataset is a curated subset of the Haemopedia RNA-Seq atlas. Human cells were from buffy coats of healthy donors, and mouse cells were from blood, bone marrow, spleen, and lymph nodes.
- Neutrophil dataset (Table 2): 195 samples (including the 9 Haemopedia neutrophil samples, Choi J 2019), 136 human samples from 13 studies, 59 mouse samples from 11 studies. All studies in this dataset were selected only to contain neutrophils. A subset of this dataset from studies with inflamed samples as well as healthy controls was used for differential expression testing and inflammatory core signature construction. Other subsets of samples from studies not selected for differential expression analysis have been used in analyses focusing on healthy control samples (Fig. 2).

### RNA-Seq of HoxB8 cells

- Khoyratty TE 2021[36] 18 samples

### ATAC-Seq

- Khoyratty TE 2021[36] 5 peak annotations

### Flow cytometry

- This study samples from 8 human donors and 9 mice
- This study samples from different organs of 6 mice

### Single-cell RNA sequencing

- Xie et al., 2020
- Montaldo et al., 2022
- Combes et al., 2021

### Data retrieval and processing

We downloaded raw sequencing reads for the selected studies to the MLS&WISO bwForCluster using release 1.5 of the nf-core[73] fetchngs pipeline and quantified them using release 3.6 of the nf-core rnaseq pipeline. The pipelines were launched using nextflow[74] (v22.04.0). To ensure high reproducibility, all pipeline processes were run inside singularity (v3.9.2) containers. For bulk RNA-seq samples, we mapped all downloaded samples using salmon[75] (v1.5.2) with the parameters libType set to 'A' and indexing the reference genomes with 21 base k-mers. Quantified transcripts were summarized to the gene level using bioconductor-tximeta[76] (v1.8.0). All human samples were mapped to the GRCh38 genome. All mouse samples were mapped to

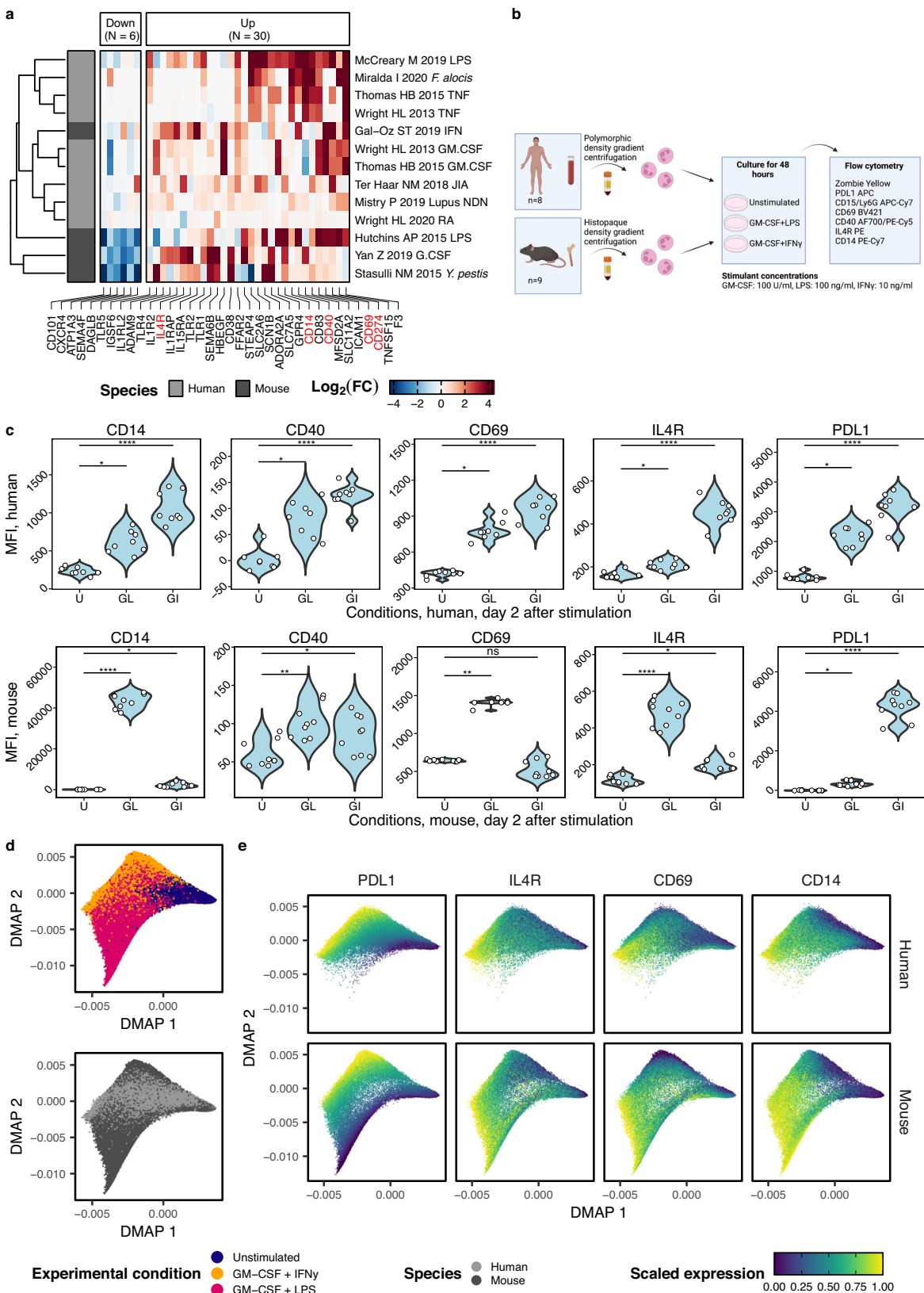

the GRCm39 genome unless stated otherwise. Quality control was conducted using FastQC (Supplementary Data 1). Author-supplied metadata was queried using GEOquery[77] (v2.64.0) and integrated manually to ensure consistency across studies (Supplementary Data 2). R (v4.2.0) was used for downstream analyses. Bioconductor (v3.15) and additional packages were used for downstream analyses

and visualizations[51,78–80]. Sequencing depths (total amount of mapped reads) for human samples in the lineage atlas dataset ranged between 11,970,326 and 16,060,356 (median 13,267,958), for mouse samples between 16,076,257 and 33,595,867 (median 18,689,418; Supplementary Fig. 2a, b). In the neutrophil dataset, sequencing depths ranged between 195,712 and 57,144,004 (median 29,741,078)

**Fig. 6 | Experimental validation of the core inflammation program on the protein level. a** Inflammation-specific protein-coding genes were identified by filtering the genes depicted in Fig. 3c for the surfaceome as previously described[45]. Genes that encode proteins selected for validation (based on antibody availability and panel design) are labeled in red. **b** Experimental overview. Human and mouse neutrophils were isolated from peripheral blood or bone marrow, respectively, cultured 48 h, and analyzed in flow cytometry. **c** Flow cytometry analysis of resting and activated mouse and human neutrophils. The gating strategy is depicted in Supplementary Fig. 11. Significance indicates adjusted *P*-values of a Dunn's test (two-sided) that followed a Kruskal-Wallis *H* test (significant for all markers). Exact *P*-values are provided in the Source Data file. **d** Top, diffusion map embedding of neutrophils from humans and mice cultured for 48 h with or without GM-CSF + LPS and GM-CSF + IFN-γ. Diffusion map embedding calculated based on CD69, CD14, IL-4R, and PD-L1. Bottom, diffusion map embedding of human and mouse neutrophils, colored by species. **e** Diffusion map embedding colored by marker expression highlights a continuum driven by increasing expression of the activation markers. Source data are provided as a Source Data file.

for human and between 1,022,486 and 42,005,334 (median 12,429,230) for mouse samples. The subset of samples that was selected for differential expression testing and inflammatory core program calculation was sequenced between 8,135,455 and 57,144,004 (median 33,522,466) for human and between 1,022,486 and 36,405,228 (median 6,388,735; Supplementary Fig. 2c, d) for mouse samples.

For single-cell RNA sequencing data, the raw sequencing reads were downloaded as described above and aligned using cellranger (v7.1.0) using the GRCh38 genome for human datasets and mm10 genome for mouse samples, respectively. Downstream analysis was carried out in Python (v3.10) using the scanpy[81,82] API (v1.9.3) for data analysis and visualization.

### Orthology analyses and mapping
For downstream analyses, genes were mapped using ENSEMBL Version 107[2]. We restricted all our composite cross-species analyses to protein-coding genes with a high-confidence orthology relationship and available gene symbols in both species. Mouse and human gene expression datasets were combined based on these orthologs.

### Identification of lineage-associated genes
Lineage-associated genes were identified using a linear model-based differential expression test, implemented in limma[83] (v3.52.0) and edgeR[84-86] (v3.38.0). Differential expression testing was restricted to protein-coding genes that could be assigned high-confidence orthologs between human and mouse samples. We constructed a cross-species count matrix based on those mappings and referred to each mapped gene by its human gene symbol. Counts were filtered using edgeR's filterByExpr filtering approach. We applied TMM normalization to account for differences in library composition. We then transformed counts to $\log_2$(CPM) values and estimated weights for each observation using voom. We applied limma to fit a linear model to our data and calculated differential expression for a given lineage against all remaining lineages. Lineage-associated genes were defined as genes that were differentially expressed in each lineage against all other lineages at a Benjamini-Hochberg corrected *P*-value of ≤0.05 and a $\log_2$ FC >0. Genes were ranked according to their F statistic, and up to 200 genes were selected per lineage.

### Lineage PCA, correlation analysis, and clustering
We used these balanced lineage-associated gene sets to perform PCA as well as correlation and clustering analysis on all samples. Human and mouse samples were combined as described above. To emphasize our focus on comparisons between lineages, we mean-centered $\log_2$(CPM) for each species prior to combining the count matrices. A PCA was computed for all integrated samples, taking the concatenated lineage-associated gene sets as input features. Correlation of expression analyses was performed based on the same features, calculating Pearson's *r* correlation coefficient for each inter-sample combination. We subsequently performed a hierarchical clustering analysis on the obtained correlation coefficients.

### Comparison of neutrophil lineage gene expression profiles in resting neutrophils
To compare expression patterns of neutrophil lineage-associated genes in resting human and mouse neutrophils, we first defined lineage-associated genes for human and mouse samples separately. We defined those genes as lineage-associated that were upregulated (Benjamini-Hochberg corrected *P*-value of ≤0.05 and a $\log_2$ FC >1) in neutrophils against all other lineages. We next mapped those gene sets to their human and mouse counterparts, considering only high-confidence one-to-one, one-to-many, and many-to-many orthology relationships. Based on those mappings, we merged all genes detected as lineage-specific in either of the considered species. We also included genes detected as lineage-specific in either species but could not be mapped to a high-confidence ortholog. The obtained genes were subset only to include genes that showed evidence of expression in the inflammatory dataset.

Taking the computed mappings and $\log_2$(TPM+1) expression values of mapped gene-gene pairs, we tested for differential expression of those pairs between species using a linear mixed model[87] (lme4 v1.1-29) accounting for study-related batch effects by including the study annotation as a random effect:

Full model: $\log_2$(TPM+1) ~ species + (1|study)
Null model: $\log_2$(TPM+1) ~ (1|study)

*P*-values were computed by performing a likelihood ratio test between these models. We subsequently adjusted those values using the Benjamini-Hochberg correction method based on the total number of tested gene-gene pairs (genes that appeared as lineage-specific in either species were expressed in the inflammatory dataset and could be mapped to one or more counterparts with high confidence).

Using the average expression of mapped gene-gene pairs and differential expression *P*-value, we defined 5 different expression profiles: Genes that showed high (>95th percentile of all genes that were detected as lineage-specific in either species) average expression levels in both species and did not exhibit differential expression between species (Benjamini-Hochberg corrected *P*-value ≥ 0.05, absolute beta < 1). Additionally, we defined 4 divergent clusters of genes that had high expression levels in only one of both species and showed evidence of differential expression (Benjamini-Hochberg corrected *P*-value < 0.05, absolute $\beta \geq 1$) or were abundantly expressed but could not be assigned an orthologous gene in the other species respectively.

### Differential expression testing
We performed differential expression analyses between inflamed and resting conditions on a total of 112 samples from *N* = 11 (human: 7, mouse: 4) studies. To account for potential batch effects between studies, we used DESeq2[88] (v1.36.0) in each of the studies individually to identify differentially expressed genes in inflamed compared to healthy control samples. Each study's gene list was pre-filtered to only include genes with counts >1 in at least 1 sample before differential expression analysis, based on the negative binomial distribution. To remove noise while preserving significant differences, $\log_2$ fold change results were then shrunk using the *apeglm* package[89]. Differential gene expression results were additionally filtered through DESeq2's default independent filtering approach, as well as its count outlier filtering.

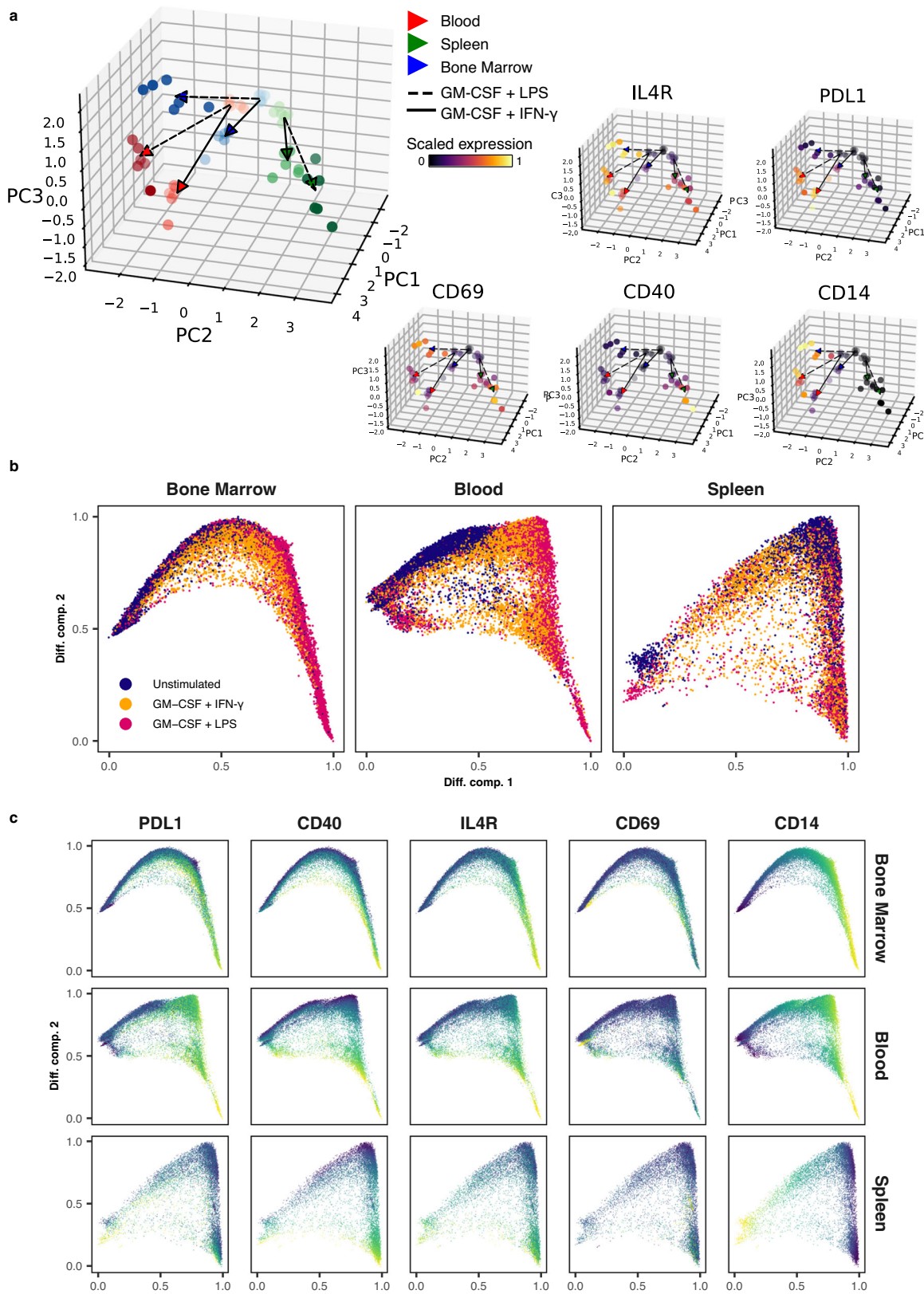

**Fig. 7 | Neutrophil origin and cytokine stimulation determine the manifestation of the core inflammation program. a** Left: Principal component analysis performed on the median marker expressions per sample reveals distinct phenotypes of neutrophils driven by organ of isolation (color coded) as well as cytokine stimulation (line-segment coded). Right: PCA plots colored by the scaled expression for each of the quantified markers. **b** Diffusion Map embedding of neutrophils from bone marrow, blood, and spleen cultured for 8 h with or without GM-CSF + LPS and GM-CSF + IFN-γ. Embeddings were calculated based on the expression levels of CD69, CD14, IL-4R, PD-L1, and CD40. **c** Diffusion Map embedding of resting and cytokine-stimulated neutrophils (as in **b**) from bone marrow, blood, and spleen, colored by marker expression. Source data are provided as a Source Data file.

**Table 1 | Lineage atlas dataset**

| Lineage | Species | N samples |
|---|---|---|
| B cells | Human | 8 |
| Dendritic cells | Human | 7 |
| Monocytes | Human | 7 |
| Neutrophils | Human | 3 |
| NK cells | Human | 5 |
| T cells | Human | 10 |
| B cells | Mouse | 2 |
| Dendritic cells | Mouse | 4 |
| Monocytes | Mouse | 6 |
| Neutrophils | Mouse | 6 |
| NK cells | Mouse | 2 |
| T cells | Mouse | 16 |

**Table 2 | Neutrophil dataset**

| Species | Study name | Used for signature construction | N samples |
|---|---|---|---|
| Human | Adrover JM 2020[10] | FALSE | 6 |
| Human | Catapano M 2020[11] | FALSE | 11 |
| Human | Choi J 2019[12] | FALSE | 3 |
| Human | Franco LM 2019[13] | FALSE | 2 |
| Human | Grabowski P 2019[14] | FALSE | 22 |
| Human | Vecchio F 2018[19] | FALSE | 8 |
| Human | McCreary M 2019 | TRUE | 10 |
| Human | Miralda I 2020[16] | TRUE | 16 |
| Human | Mistry P 2019[17] | TRUE | 28 |
| Human | Ter Haar NM 2018[107] | TRUE | 6 |
| Human | Thomas HB 2015[18] | TRUE | 12 |
| Human | Wright HL 2013[21] | TRUE | 6 |
| Human | Wright HL 2020[20] | TRUE | 6 |
| Mouse | Bhalla M 2021[22] | FALSE | 7 |
| Mouse | Casulli J 2019[23] | FALSE | 3 |
| Mouse | Coffelt SB 2015[24] | FALSE | 4 |
| Mouse | Choi J 2019[12] | FALSE | 6 |
| Mouse | Germann M 2020[26] | FALSE | 4 |
| Mouse | Hsu BE 2019[27] | FALSE | 4 |
| Mouse | Zhu YP 2018[31] | FALSE | 3 |
| Mouse | Gal-Oz ST 2019[25] | TRUE | 12 |
| Mouse | Hutchins AP 2015[28] | TRUE | 4 |
| Mouse | Stasulli NM 2015[29] | TRUE | 6 |
| Mouse | Yan Z 2019[30] | TRUE | 6 |

## Identification of a core inflammation program

To assess which inflammation-driven changes in the neutrophil transcriptome are shared across conditions and conserved across species, we applied a Fisher's combined test to the adjusted $P$-values of each gene in each study, restricting the analysis to genes that passed our expression filter as well as DESeq2 filters in ≥80% of studies. This analysis provided a Benjamini-Hochberg-corrected composite $P$-value for all genes and allowed us to rank genes by their likelihood of dysregulation in inflammation. Additionally, we calculated a mean $\log_2$ fold change for each gene across all studies.

Based on a rank-$P$-value plot (Fig. 3b), we determined a $P$-value cutoff at a rank equaling 500, corresponding to an adjusted Fisher $P \leq 6.164117 \times 10^{-40}$. Genes with an absolute $\log_2$ fold change greater than or equal to 0.5 and an adjusted $P$-value below our defined threshold were considered conserved in inflammation. We defined the upregulated subset ($\log_2 FC \geq 0.5$) of those conserved genes as the core inflammation program.

## Pathway enrichment analysis

Inflammatory pathway enrichment analysis was performed for each study individually using the fGSEA implementation of the Gene Set Enrichment Analysis method. For each study, the differential expression analysis results were ranked by $\log_2$ fold change. Enrichment was calculated for hallmark gene sets that were retrieved from the Molecular Signatures Database[90] (v7.5.1).

## Data preprocessing for single-cell RNA-seq

Datasets were imported using the raw count matrices from cellranger. First, empty droplets were determined by estimating the profile of ambient mRNA and testing deviations from this profile using a Dirichlet-multinomial model of UMI count sampling as implemented in the EmptyDrops method[91] (implemented in the DropletUtils package, v1.18.1). Ambient RNA correction was applied using the soupX-algorithm[92] (v1.6.2), and doublets were determined using a computational doublet detection tool that uses artificially created cell doublets to identify real cell doublets by nearest-neighbor-analysis in gene expression space[93] (v1.12.0). Cells expressing hemoglobin-related genes in a proportion above 0.02 were excluded, as well as cells containing less than 250 (Xie et al., Montaldo et al.) or 100 (Combes et al.) unique genes per cell. Cells with a content above 5% (Xie et al.) or 10% (Combes et al., Montaldo et al.) of mitochondrial genes were also excluded. Cell types were identified using the SingleR package[94] (v2.0.0) in R with BlueprintEncodeData and MonacoImmuneData as reference datasets for human datasets and ImmGenData as reference dataset for mouse data, as provided by the package celldex[94] (v1.8.0). Data were log-normalized, and neutrophils were selected. For UMAP visualization, the 2000 genes containing the highest variance were selected and UMAP was computed using the scanpy API with default settings.

## Gene set enrichment in single-cell RNA-seq

The gene counts were normalized and log1p transformed using scanpy. The enrichment of genes belonging to the core program was quantified using the difference between the average expression of the core inflammation program genes and the average expression of a random reference set of genes[95] that have been sampled to match the expression distribution of the core inflammation program (scanpy score_genes function with default settings). We tested for enrichment in inflamed conditions using a linear mixed model with the sample (and organ for the dataset from Xie et al.) as a random effect and the respective treatment as a fixed effect. For datasets with multiple inflamed conditions, a respective model was created for each comparison separately. $P$-values were calculated by performing a maximum likelihood test between both models as described above. For visualization, the gene set scores were quantile-capped to the 5th and 95th percentile.

## Transcription factor enrichment analysis

In order to identify regulators associated with genes induced or downregulated in inflammation, we used ChEA3[96] with default settings as described[9], using the 250 most significantly up- and downregulated genes, respectively, for each condition, ranked by their adjusted $P$-value. We then calculated the arithmetic mean of the negative logarithms of the ChEA3 scores per species and transcription factor to compare average TF activity across species.

We used a paired t-test to assess significant differences between a TFs ChEA3-enrichment in up- against downregulated genes across all comparisons. The resulting $P$-values were corrected using a Benjamini-Hochberg correction for all tested TFs. We used these corrected $P$-values to determine if a TF was significantly more enriched in genes

upregulated in inflammation or vice-versa. We subsequently inferred transcription factor activity using DoRothEA[97] (v1.8.0) and decoupleR[98] (v2.2.2), taking advantage of the species-specific transcription factor databases. Here, $\log_2$ fold change matrices per species served as input, leading to enrichment scores with their respective *P*-values. For downstream analyses, we calculated the mean enrichment scores per species and preserved the highest observed *P*-value for each transcription factor. For data visualization (Fig. 5a), only transcription factors where the respective gene had an assigned *P*-value in Fisher's combined test are shown (*N* = 680). Genes encoding transcription factors with a mean $\log_2$ fold change ≥0 were merged with transcription factor scores that were derived from the upregulated score set (as in Supplementary Fig. 8b), and genes with a mean log2 fold change <0 were merged with transcription factor scores that were derived from the downregulated score set (as in Supplementary Fig. 8a). Labeling was selectively applied for genes encoding transcription factors with the 10 highest absolute $\log_2$ fold changes per direction which fall under the adjusted Fisher's combined *P*-rank cutoff of 500 genes, as well as the 5 transcription factors with the lowest *P*-values and *CEBPB*.

### ATAC-sequencing analysis

We retrieved ATAC-sequencing data from mice that were subjected to the air pouch model of acute inflammation (GEO: GSE161765, mapped to the GRCm38 genome). Genes annotated based on differentially accessible peaks as defined in the study ($P_{adj} < 0.05$, $\log_2$ fold change > 1.5) were compared with the conserved upregulated genes as defined in the core inflammation program. The ratio and number of core inflammation program genes that were associated with projected increased accessibility served as an input for pairwise Fisher's exact tests, and *P*-values were adjusted using the Benjamini-Hochberg method. For each comparison, the significance of the number of core inflammation genes with increased accessibility was retrieved by comparing the number with the results of this analysis using a 1000-fold repeated random selection of expression-matched background genes (as described below for RNA sequencing).

### RNA-sequencing analysis of zymosan-treated HoxB8 cells

We retrieved featureCounts (per ENSEMBL-ID) from HoxB8 cells that were subjected to differentiation and zymosan treatment (GEO: GSE161765, mapped to the GRCm38 genome). Differential expression analysis was performed as described in the respective section above. We restricted the analysis to HoxB8 cells that were differentiated for 5 days and then compared (1) wild-type cells that were treated for 2 h with zymosan (50 µg/ml) or DMSO (control), (2) resting stable knockout HoxB8 cell lines versus wild type, (3) zymosan-treated stable knockout HoxB8 cell lines versus zymosan-treated wild-type HoxB8 cells. A significant up- or downregulation of core inflammation program genes was then assessed by performing pairwise overrepresentation analyses. For each overrepresentation analysis, we defined differentially expressed genes as genes with an *FDR* ≤ 0.05 and a |log2 fold change| ≥ 1. We then used goseq (v1.48.0) to calculate a Probability Weighting Function for the given set of genes, and calculated *P*-values by approximating the true distribution by the Wallenius non-central hypergeometric distribution as previously described[99].

The control expression was calculated as previously described[95]. For each comparison, the gene expression was distributed in 25 bins. Then, each core inflammation program member was assigned to its respective bin. The randomized sets were then sampled according to the distribution of core inflammation program gene expressions. This sampling was repeated 1000 times.

### Experimental validation

The list of ranked conserved inflammatory response genes was filtered to include genes encoding surface proteins using the surfaceome resource[45]. The remaining *N* = 69 surface protein-encoding genes were then filtered by available human and mouse antibodies (BioLegend), and a panel consisting of CD14, CD69, CD40, IL-4R, and PD-L1 was selected for validation.

### Human samples

Neutrophils were isolated using density gradient centrifugation with Polymorphprep as previously described[8]: 30 ml whole blood was layered onto 20 ml Polymorphprep (Progen #1114683) and centrifuged at $535 \times g$ for 35 min. The PBMC-containing layer was discarded by suction, and neutrophils were recovered and subjected to hypotonic lysis using 0.2% NaCl. The cells were subsequently washed with cell culture medium (RPMI 1640 (Gibco #21875-034)) supplemented with 10% heat-inactivated FBS (PAN Biotech #3302/P101102) and 1% Gluta-MAX (Gibco #35050-061) and seeded at 5 million cells per 6 wells in a total volume of 5 ml at a humidified atmosphere at 37 °C with 5% $CO_2$. The cells were cultured over 48 h either in the absence of cytokines (vehicle control), with GM-CSF + IFN-γ or GM-CSF + LPS. GM-CSF was used at a final concentration of 100 U/ml (R&D #215GM), IFN-γ at 10 ng/ml (BioLegend #570208), and LPS at 100 ng/ml (Invivogen #tlrl-3pelps). After 48 h, 1 million cells were collected and stained using the Zombie Yellow Fixable Viability Kit (BioLegend #423103) for live/dead discrimination, followed by an antibody panel (Supplementary Table 1) in 50 µl of FACS buffer (2% FBS, 5 mM EDTA and 0.1 sodium azide in PBS) for 25 min.

### Mouse samples

Mice were housed under SPF conditions with a 12 h light/dark cycle, a humidity of 50–60%, a temperature of 22 ± 2 °C and food and water available ad libitum. Male and female C57BL/6J mice were sacrificed by cervical dislocation, and bone marrow was extracted by flushing with RPMI. Neutrophils were enriched by density centrifugation using Histopaque 1077 (Sigma-Aldrich #10771) and Histopaque 1119 (Sigma-Aldrich #11191). Cells were recovered from the interphase of both Histopaque layers and centrifuged. Cells were washed with RPMI containing 10% FBS and 1% Glutamax and seeded at $10^6$ cells/ml in 48 well plates in a total volume of 500 µl. Mouse GM-CSF (Peprotech #315-03, 100 U/ml), mouse IFN-γ (Peprotech #315-05, 10 ng/ml), and LPS (Invivogen #tlrl-3pelps, 100 ng/ml) were added to the medium for 24 h and 48 h in combination as indicated in the respective figures. Cells cultured in the absence of cytokines were used as controls. After the indicated times, cells were collected and stained with the antibody panel (Supplementary Table 2) in 50 µl of FACS buffer containing 2% FBS, 5 mM EDTA and 0.1% sodium azide.

To assess neutrophils from different organs, mice were sacrificed by cardiac puncture under generalized anesthesia. Subsequently, the femora and tibiae were flushed with PBS to obtain bone marrow. Any remaining fat was removed from spleens, and splenic tissue was mechanically disintegrated using the back of a syringe. Cells were pelleted at $400 \times g$, and erythrocytes were lysed using ACK Lysing Buffer (Lonza #10-548E) for 5 min at 4 °C. Cells were seeded at $10^6$ cells/ml in 48 well plates in a total volume of 500 µl. Cytokines were added as described above for a total of 8 h.

### Flow cytometry

Flow cytometry was performed on a BD LSRII flow cytometer. At least 50,000 events were recorded per sample. FCS files were exported by FACSDiva and subsequently gated and compensated in FlowJo (v10.8.0) for single, living, and CD15[+] (human) and Ly6G[+] (mouse) cells. Eosinophils were excluded based on high autofluorescence in the live/dead (Pacific Orange) channel (Supplementary Fig. 11). Gated events and their median fluorescence intensity values were exported and concatenated into a single-cell experiment using CATALYST[100] (v1.16.2) in R (v4.2.0). The dataset was arcsinh transformed using manually determined cofactors[8,101] and clustered by FlowSOM clustering and Consensus-Plus-Metaclustering. For combined analysis of human and

mouse cells, both datasets were mean-centered, scaled, and combined into one SingleCellExperiment. Dimensionality reduction was performed using the DiffusionMap algorithm as implemented in the CATALYST package with standard settings. For visualization, a random subset of 1500 cells per sample was plotted using ggplot2[102] (v3.3.5). Principal components were calculated based on the median fluorescence values of the respective marker proteins per sample and plotted using matplotlib (v3.5.1) in Python (v3.9.1).

## Gene ontology enrichment analysis
We used EnrichR[103–105] to assess enriched gene sets in a given list of genes. The analysis was restricted to terms annotated in GO_Biological_Process_2021.

## Linear mixed-effect model
To validate the core inflammation program derived from Fisher's combined test, we globally tested for differential expression between resting and inflamed cells including all samples used in the Fisher's combined testing approach. We accounted for batch effects by correcting gene counts for the study using ComBat-Seq[106] (sva v3.44.0). From batch-corrected counts, we calculated TMM-normalized $\log_2$ counts per million that were subsequently quantile normalized and then used as input for linear modeling. Modeling was implemented using lme4[87] (v1.1-29) to fit a linear mixed-effects model (LMM) to normalized counts. The linear formulae we fit for each gene were defined as $full$ : $expression \sim condition + 1|study$ and $reduced$ : $expression \sim 1|study$, where the variable to test for was $condition$, and the $study$ was used as the covariate that was considered to be the random effect. We retrieved $\beta$ as an estimate for the $\log_2$(FC) from the full model and subsequently performed a likelihood ratio test to compare the $full$ with the $reduced$ model and to retrieve the respective $P$-values. $P$-values were then adjusted using the Benjamini-Hochberg procedure. We accounted for batch effects by correcting gene counts using ComBat-Seq[106] (sva v3.44.0).

## $\pi_1$-statistic
Using the qvalue-package[39,50,51] (v2.28.0), we calculated the $\pi_1$-statistic (1- $\pi_0$) as an estimated proportion of truly significantly differentially expressed genes for a given set of $P$-values. To account for a selection of genes potentially biased toward low $P$-values when testing for the replicability of DEGs between studies, qvalue-calculation was implemented via the qvalue_truncp function.

## Gene expression modules using WGCNA
For WGCNA[39] analysis, we selected the same samples that were used for differential expression testing. We accounted for batch effects by correcting gene counts using ComBat-Seq[106] (sva v3.44.0). From batch-corrected counts, we calculated TMM-normalized $\log_2$ counts per million that were subsequently quantile normalized and then used as input for WGCNA. The network was constructed as a signed network, using a soft thresholding power of 13, a minimum module size of 30, and a merge cut height of 0.25. Modules with more than 1000 genes were removed from subsequent analyses.

## Statistical analyses
Correlations indicated on scatter plots represent Pearson's $R$ (Pearson's correlation coefficient) with their respective $P$-value. For comparisons of the mean, we used the Mann-Whitney $U$ test (two groups) or Kruskal-Wallis $H$ test (three groups) if the Shapiro-Wilk test indicated non-normality in at least one group. When multiple comparisons were performed, $P$-values and/or asterisks indicate adjusted $P$-values using the Holm-Bonferroni method unless stated otherwise. For pairwise comparisons, the Mann-Whitney $U$ test was used post-hoc, taking multiple comparisons into account using the Holm-Bonferroni method. To test for categorical associations, we used Fisher's exact test. Asterisks

represent the following $P$-value ranges: $P > 0.05$, ns. $0.01 < P \le 0.05$, *. $0.001 < P \le 0.01$, **. $0.0001 < P \le 0.001$, ***. $P \le 0.0001$, ****.

## Reporting summary
Further information on research design is available in the Nature Portfolio Reporting Summary linked to this article.

## Data availability
All sequencing data was used from publicly available platforms (Gene Expression Omnibus and European Nucleotide Archive), and all accession numbers are listed in Supplementary Data 2. Flow cytometry data have been deposited at flowrepository.org under the accession FR-FCM-Z6U3 and FR-FCM-Z6U4. All other data are available in the article and its Supplementary files or from the corresponding author upon request. Source data are provided with this paper.

## Code availability
Analysis code is publicly available on GitHub: https://github.com/rgb-lab/inflamed_neutrophil_transcriptome.

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

## Acknowledgements

N.S.H. and F.A.R. were supported by MD fellowships from the Boehringer Ingelheim Fonds. F.A.R. and T.E. were supported by an MD/PhD fellowship from the Medical Faculty of Heidelberg. P.A.N. was supported by NIH grants R01AR065538 and P30AR070253. This work was supported by grants to R.G.-B. from Deutsche Forschungsgemeinschaft (DFG, GR 5979/2-1, 517717827), Else Kröner-Fresenius-Stiftung (2022_EKEA.72), state of Baden-Wuerttemberg within the Centers for Personalized Medicine Baden-Wuerttemberg (ZPM) and a research grant from the German Society for Rheumatology (DGRh). Figure 6b and Supplementary Fig. 1 were created with BioRender.com. The authors acknowledge support by the state of Baden-Württemberg through bwHPC and the German Research Foundation (DFG) through grant INST 35/1597-1 FUGG.

## Author contributions

N.S.H. and F.A.R. conceptualized and designed the study, performed computational and statistical analysis, provided conceptual input in experimental planning, analyzed experiments, conceptualized the core inflammation program, and wrote the manuscript. T.E. performed computational and statistical analysis, planned experiments, processed human and mouse samples, performed and analyzed flow cytometry, and wrote the manuscript. H.-M.L., C.M.-T., P.A.N. and G.W. guided data analysis, interpretation, and validation. R.G.-B. conceptualized and designed the study, performed computational and statistical analysis, provided conceptual input in experimental planning, analyzed experiments, conceptualized the core inflammation program, supervised the entire project, and wrote the manuscript.

## Funding

## Competing interests

The authors declare no competing interests.

## Additional information

[1]Division of Rheumatology, Department of Medicine V, Heidelberg University Hospital, Heidelberg, Germany. [2]Institute for Immunology, Heidelberg University Hospital, Heidelberg, Germany. [3]Division of Immunology, Boston Children's Hospital, Harvard Medical School, Boston, MA, USA. [4]Broad Institute of MIT and Harvard, Cambridge, MA, USA. [5]Division of Rheumatology, Inflammation and Immunity, Brigham and Women's Hospital, Harvard Medical School, Boston, MA, USA. [6]MRC Molecular Haematology Unit, MRC Weatherall Institute of Molecular Medicine, University of Oxford, Oxford, UK. [7]Oxford Centre for Haematology Unit, MRC Weatherall Institute of Molecular Medicine, University of Oxford, Oxford, UK. [8]Department of Medicine V, Hematology, Oncology and Rheumatology, Heidelberg University Hospital, Heidelberg, Germany. [9]Molecular Medicine Partnership Unit, European Molecular Biology Laboratory (EMBL), University of Heidelberg, Heidelberg, Germany. [10]Deutsches Zentrum für Immuntherapie (DZI), Friedrich Alexander Universität Erlangen-Nürnberg and Universitätsklinikum Erlangen, Erlangen, Germany. [11]Department of Internal Medicine 3 – Rheumatology and Immunology, Friedrich Alexander Universität Erlangen-Nürnberg and Universitätsklinikum Erlangen, Erlangen, Germany. [12]These authors contributed equally: Nicolaj S. Hackert, Felix A. Radtke, Tarik Exner. ✉e-mail: ricardo.grieshaber@fau.de

