## [Peer Review File · Nature Communications]

Human and mouse neutrophils share core transcriptional programs in both homeostatic and inflamed contextsREVIEWER COMMENTS

Reviewer #1 (Remarks to the Author):

The present manuscript entitled « Human and murine neutrophils share core transcriptional programs in both homeostatic and inflamed contexts » by the team of Dr Grieshaber-Bouyer attempted to highlight similarities and differences of gene expression program between murine and human neutrophils.

This study follows a previous study showing that neutrophil heterogeneity is driven by a chronological sequence of maturation and activation gene expression that was called neutrotime (Grieshaber-Bouyer et al, 2021).

The data reported in this manuscript are based mainly on analysis on publicly available RNA sequencing data and also on new experimental data using an in vitro model of murine neutrophil differentiation.

The authors compare the transcriptome of neutrophils between murine and human datasets, between healthy controls and patients with different inflammatory diseases, between an activated and resting state as well as between blood and different tissues.

They showed that from all leukocytes neutrophils had the highest correlation of gene expression between humans and mice.

They confirmed their data on a protein level with surface markers on activated neutrophils that belong to the core inflammatory genes. Further, they confirmed increased chromatin accessibility of core inflammation genes in the activated and tissue neutrophils.

This manuscript is providing relevant information in the field of neutrophil research because it allows to compare different sets of data and help the reader to have an overview on different studies focused on the molecular mechanisms that regulate neutrophil resting versus activated state.

This manuscript also highlights essential pathways that are activated whatever the

activation stimulus as a core program which seems to be similar in mice and humans.

I have basically two types of remarks: firstly, I have some technical remarks on the way to present and explain the complex data; secondly, I would like to highlight some more fundamental issues raised by the conclusion that murine neutrophils would be excellent models for human neutrophils. I think this is still an open question and this assertion might be misleading in some conditions. This latter point could improve the quality and the significance of this interesting manuscript.

Technical points

1) Brief explanation and summary of the different techniques

Although the manuscript is well-written and the supplementary data are useful, the presented data are not easily understandable for people not familiar with ATAC-sequencing data and with WCGNA when they read the main manuscript (without consulting the supplemental files). Maybe the authors might explain briefly in one sentence and expand these acronyms.

2) Figure 1 is very complicated and might be simplified.

I would suggest to remove panel A which seems to be a graphical representation of the study. This can be included in a graphical abstract.

Panel 1B could have the legend included (instead to have the legend as the last panel of the figure).

In panel C, it is not clear what the numbers mean ? Are they the number of experiments for each cell type because of the different colors ?

Is panel E related only to blood neutrophils for both human and mice or are the data from mice derived from bone marrow neutrophils ?

3) Definition of the resting versus activated state.

It would be helpful for the reader if the authors could explain in the beginning how they distinguished between resting and activated neutrophils (eg. For figure 1 and 2 page 4).

Can the authors explain what they mean with individual samples on page 6 ? Did they take a new group of patients and healthy controls to confirm their finding of the 179 core inflammation genes?

4) Can the authors explain how they chose the two transcription factors JunB and Cebpbeta for their knock out model using the HoxB8 out of the top transcription factors with the highest predicted activity?

5) Identification of a core program between mice and human does not mean that the pathophysiological mechanisms are similar between murine models and human diseases

The authors found a high conservation of lineage-specific gene, particularly the ones whose refers to neutrophils. According to the integrative analysis of transcriptomes, they could define a core inflammation program relatively well- preserved between mice and humans. Later they validated this program in different experimental models.

Although the authors highlight a restricted number of common and conserved molecular effectors between mice and human, the conclusion that the neutrophil activation would rely on similar pathogenic mechanisms is misleading. Most likely, the mechanisms that would be relevant for a given pathology would be outside of this core program which is, by definition not specific at all.

Although this analysis is useful and well done, the data that are presented should not be overinterpreted. The authors should clearly explain the intrinsic differences between murine and human neutrophils related to infectious diseases for example. The authors mention the lack of defensins in murine neutrophils but there are many differences due to the co-evolution of pathogens with the host in term of immune adaptation.

The authors could refer to a previously published papers. For instance the article by W. Nauseef in Immunol Review (DOI: 10.1111/imr.13154) has a complete review of the literature on this topic and clearly highlights the differences between murine and human neutrophils. In the conclusion, the authors explain how both human and murine systems

could be complementary in neutrophil studies provided that some cautions are taken.

The discussion of the present manuscript could be enriched by these elements which should be taken into consideration for any researcher working on neutrophils.

Reviewer #2 (Remarks to the Author):

Although the transcriptional profiles of neutrophils in normal and pathophysiological conditions in mice and humans have been demonstrated separately in recent studies, in context of comparing these profiles among mice and humans to identify the core features that are shared between these two species in different conditions is informative. The findings are interesting, and the paper is well-written. However, in addition to demonstrating the general transcriptomic profiles and pathways analyzed, it would be interesting to have revealed the effector molecules (which promote inflammation, and tissue injury) of the neutrophils among the groups they included in their study. For example, ROS levels and NETosis.

Aged neutrophil markers are CXCR4+ and CD62L (L-selectin) low/negative; however, in Supplementary Figure 7, flow cytometry analysis of neutrophil aging in vitro, there is no information regarding these markers.

What were the criteria for including and excluding the publicly available data sets that the authors analyzed in their manuscript? Per Fig 3A, the authors have mentioned the stimuli or disease conditions of the data they studied. Still, the question is why the authors included Lupus and JIA but not the bacterial infection models or the clinical samples of sepsis.

A typographical error: Fig 5D in the result's section statistics (0.79) and Fig 5D statistics (0.0011) are not matched well.

Reviewer #3 (Remarks to the Author):

Grieshaber-Bouyer and colleagues present a re-analysis of neutrophil transcriptomic datasets in both mice and human. The goal is to identify and characterize conserved core signatures both at baseline (resting) and after neutrophil activation following different pro-inflammatory stimuli.

The authors first identify lineage-defining genes and then build classes of genes based on the concordance of their expression in both species. Using a panel of existing studies, the authors perform differential expression between resting and activated neutrophils to compile a core inflammatory signature (179 genes), arguably conserved across stimuli and species. Further data integration is performed using linear models and co-expression network analysis. Additional computational analyses are carried out to validate the parallelism between gene regulatory programs in human and mice using transcription factor enrichment and activity analysis, along with chromatin accessibility data from a specific model of acute inflammation. The regulome of 2 of the identified factors (JUNB and CEBPB) is computationally characterized from public expression data generated on ex-vivo neutrophils knock-out experiments. Flow cytometry is then applied to address how the core transcriptional inflammatory signature is recapitulated at the protein level, again revealing both conserved and species-specific features. Finally, the tissue-specificity of the core program is evaluated using stimulation experiments and flow cytometry for a small panel of selected genes.

In its current form, and although the re-analysis effort constitutes an interesting meta-analysis, it is this referee's opinion that the results and conclusions of this work largely rely on previous studies consisting on a small number of samples and limited diversity, while the scope of the study is definitely ambitious, comprising human, mouse and diverse pro-inflammatory signals. The identification of a conserved, universal baseline and inflammatory core transcriptional signature in neutrophils would likely require a properly designed dataset, and its comprehensive characterization and validation at the chromatin, protein and regulatory level should rely on a properly matched multi-omic screen. In addition, a more detailed discussion, comparative analysis or integration with relevant studies reporting fine-grained neutrophil transcriptional signatures is lacking.

My major concerns are summarized below, where I focus on analytical and statistical details and how they relate with the major conclusions drawn by the authors:

- Neutrophils are characterized by a relatively less complex transcriptome as compared to other immune subsets. Other factors specific to this population (high RNase content, cell loss during library preparation) are known to affect the quality of neutrophil

transcriptomics. These factors can determine the suitability of some of the analytical choices but are not discussed by the authors. For instance, the higher correlation between mouse and human lineage-defining genes as compared to other immune subsets could be due a lower transcriptome complexity, reflected in a higher segregation between low and highly expressed genes in neutrophils. As lineage-defining genes are likely to be on the high-expression tail of the distribution, this could inflate the correlation across species for these genes. Similarly, any randomization analysis (figure 5D) should account for this using matched overall expression levels between target and random sets. Otherwise, random sets might be largely constituted by lowly expressed genes, artificially biasing the differential enrichment results.

- On a related note, the identification of lineage-defining genes is based on a very small number of neutrophil samples (3 for human, 6 for mouse). As no additional details about sequencing factors (e.g. total sequencing depth and library quality) are provided, a discussion or more in-depth evaluation of the robustness of the identified signatures would be required. And on a technical note, k-mer based RNA-Seq methodologies like the one used by the authors, have been reported to be less accurate for lowly expressed genes, potentially aggravating the effect of low-transcriptome complexity on the results. And being a transcriptome-based approach, further data loss can be expected due to a well-known prevalence of intron retention events in neutrophils. Unfortunately, “technical” factors cannot be avoided but need to be taken into account and acknowledged, particularly when they are specific to the cell type of interest.

- Similarly, most of the public data compiled and re-analyzed by the authors in the context of inflammatory responses is based on very small case-control studies (figure 3). I understand the argument to restrict the analysis to studies with both case and control data available. But the combination of the various factors (species+model+tissue+stimuli) added to the limited number of replicates per experiment severely limits the scope of what can be done. Again, no information about data quality factors is included (total depth/reads in each experiment?), and how this affects summary statistics. The authors make the correct choice on building summary statistics of dataset-specific differential expression results, but the universality of the reported results is doubtful in my opinion (Figure 3C), showing limited or inconsistent up-regulation of many “core” inflammatory genes across datasets.

Supplementary Figure 2 is also concerning given the variability in the reported numbers of

differentially expressed genes across different studies (even with similar stimuli), and the significance of the identified core. Examples are Supp. Figure 2B showing a significant number of core inflammation genes that are exclusively regulated in one species; the lack of reproducibility shown for the down-regulation of CD101 in Supp. Fig. 2C (with no effect size in most samples), similar to the aforementioned in the core program shown in Figure 3C; the relatively low and scattered GSEA enrichment scores in figure 3E across datasets.

- Regarding the attempted global analyses across studies, several methodological choices are not explained, and the results insufficiently interpreted:
 - o Why do the authors perform linear modeling based on $\log_2(\text{TPM}+1)$ scaled data? TPM is a within library normalization and therefore strongly affected by library-specific factors that may not be completely, or even partially corrected by a random term in the model. Did the authors try batch-corrected data, or between-samples variance stabilization (e.g from DESeq)? I'd say that a majority of core inflammatory genes are not identified as significant in Supp. Figure 3 (reported is 49 genes out of 179, but many other genes not in the core set show similar statistics, this needs to be discussed).
 - o WGCNA results highlights some of my concerns above if my interpretation is correct. From Supp. Figure 5, sample-specific effects are apparent from the stripes of highly regulated genes in inflammatory conditions. This is expected but it can definitely affect the analysis. No information about the concordance across replicates, which is not apparent from that figure, is provided. Modules 33 and 34 are virtually identical, and many genes in these modules are downregulated in mouse inflammatory neutrophils, with opposite trend for most genes in human neutrophils. Module 10 comprises two subsets of genes with up and down-regulation after inflammation, which questions the concept of co-expression within a given module.

- Regulatory analyses:
 - o Accessibility analysis: ATAC-Seq data is limited and was obtained from a very specific model of acute inflammation. How appropriate this data is to validate a signature obtained from a very heterogeneous, across-species dataset is uncertain and not discussed. In addition, the significance of the numbers provided in page 7 (lines 252-258) is unclear without a proper evaluation of the overall, genome-wide changes in accessibility. The

authors seem to rely on the results reported in the original study to perform some intersection analysis. In general, it is my opinion that the scope and relevance of the ATAC-seq data is very limited.

o Everything is focused on up-regulation in inflammatory conditions, which is understandable. Are there any repressive programs after activation/migration the authors can comment on?

- Given the extensive literature on the transcriptional programs in neutrophils, and the numerous recent studies using single-cell approaches to unveil the heterogeneity of this cell type, I'd expect a detailed comparative analyses between the summary results obtained in this meta-study and existing knowledge in the field. Instead, the discussion is limited to an enumeration of the analyses and results of this work. I am left wondering about the novelty of the findings reported here, the relevance and novelty of the transcription factor analyses results, how the core program identified here relates with the observed heterogeneity of human and mouse neutrophils, or even how their protein marker results could be interpreted (or even integrated somehow) with existing single-cell multi-omic datasets (CITE-Seq). Unfortunately, the lack of an in-depth evaluation of any novel findings as compared to a recapitulation of existing knowledge adds to the limited scope of the computational results.

Response letter

Reviewer #1 (Remarks to the Author):

The present manuscript entitled « Human and murine neutrophils share core transcriptional programs in both homeostatic and inflamed contexts » by the team of Dr Grieshaber-Bouyer attempted to highlight similarities and differences of gene expression program between murine and human neutrophils.

This study follows a previous study showing that neutrophil heterogeneity is driven by a chronological sequence of maturation and activation gene expression that was called neutrotime (Grieshaber-Bouyer et al, 2021).

The data reported in this manuscript are based mainly on analysis on publicly available RNA sequencing data and also on new experimental data using an in vitro model of murine neutrophil differentiation.

The authors compare the transcriptome of neutrophils between murine and human datasets, between healthy controls and patients with different inflammatory diseases, between an activated and resting state as well as between blood and different tissues.

They showed that from all leukocytes neutrophils had the highest correlation of gene expression between humans and mice.

They confirmed their data on a protein level with surface markers on activated neutrophils that belong to the core inflammatory genes. Further, they confirmed increased chromatin accessibility of core inflammation genes in the activated and tissue neutrophils.

This manuscript is providing relevant information in the field of neutrophil research because it allows to compare different sets of data and help the reader to have an overview on different studies focused on the molecular mechanisms that regulate neutrophil resting versus activated state.

This manuscript also highlights essential pathways that are activated whatever the activation stimulus as a core program which seems to be similar in mice and humans.

I have basically two types of remarks: firstly, I have some technical remarks on the way to present and explain the complex data; secondly, I would like to highlight some more fundamental issues raised by the conclusion that murine neutrophils would be excellent models for human neutrophils. I think this is still an open question and this assertion might be misleading in some conditions. This latter point could improve the quality and the significance of this interesting manuscript.

Response: We thank the reviewer for their positive assessment of the significance of our manuscript. In this revised version, we have added the required technical remarks to improve understanding of the data and updated several figures as explained below. In the discussion, we now place additional emphasis on the point that there are fundamental differences between murine and human neutrophils. We also include additional recent literature relevant for the question at hand, including single-cell data .

Technical points

1) Brief explanation and summary of the different techniques

Although the manuscript is well-written and the supplementary data are useful, the presented data are not easily understandable for people not familiar with ATAC-sequencing data and with WCGNA when they read the main manuscript (without consulting the supplemental files). Maybe the authors might explain briefly in one sentence and expand these acronyms.

Response: We expanded the acronyms (Assay for Transposase-Accessible Chromatin using sequencing; Weighted Gene Co-expression Network Analysis) and added additional explanations about these methods to the manuscript in the respective sections:

- Results section **“Migration into tissue and activation significantly enhance chromatin accessibility and expression of core inflammation genes”**:
“To test this hypothesis, we analyzed chromatin accessibility data derived from bone marrow, blood and an air pouch model of acute inflammation. This data was generated using assay for transposase-accessible chromatin with sequencing (ATAC-Seq), a method which tests genome wide chromatin accessibility. Briefly, ATAC-seq allows the analysis of chromatin accessibility by sequencing DNA fragments that are bound by a hyperactive Tn5 transposase, which preferentially inserts sequencing adaptors into open chromatin regions.”
- Results section **“The core inflammation program is detectable using different analytical strategies and detectable in single-cell data”**:
“As additional analytical approach, we performed a weighted gene co-expression network analysis (WGCNA). WGCNA constructs correlation networks and can help to identify clusters of genes (“modules”) that are co-expressed across different conditions.”

2) Figure 1 is very complicated and might be simplified.

I would suggest to remove panel A which seems to be a graphical representation of the study. This can be included in a graphical abstract.

Panel 1B could have the legend included (instead to have the legend as the last panel of the figure).

In panel C, it is not clear what the numbers mean ? Are they the number of experiments for each cell type because of the different colors ?

Is panel E related only to blood neutrophils for both human and mice or are the data from mice derived from bone marrow neutrophils ?

Response: We revised Figure 1 as suggested by the reviewer:

- Panel A is now part of the overview of the studied datasets in Supplementary Figure 1 (as *Nature Communications* does not allow graphical abstracts).
- The legend has been included in former panel b (now Figure 1a).
- The numbers in panel c are the number of genes for each lineage included in this mapping. Up to 200 lineage-associated genes with a positive \log_2 fold change and compared to every other lineage were allowed. For some lineages, less than 200 genes matched these criteria. We added an explanation to the legend.

- Panel e included also data from bone marrow neutrophils, as these were present in the haemopedia atlas. This is now explained in the methods: “Human cells were from buffy coats of healthy donors and mouse cells were from blood, bone marrow, spleen and lymph nodes.” Thus, murine but not human neutrophil samples also contained bone marrow neutrophils. Despite known maturation differences between blood and bone marrow neutrophils, the overall correlation of gene expression between species was still high.

3) Definition of the resting versus activated state.

It would be helpful for the reader if the authors could explain in the beginning how they distinguished between resting and activated neutrophils (eg. For figure 1 and 2 page 4).

Can the authors explain what they mean with individual samples on page 6 ? Did they take a new group of patients and healthy controls to confirm their finding of the 179 core inflammation genes?

Response: We defined resting neutrophils as those isolated from blood or tissue in absence of any form of disease or experimental manipulation. Activated neutrophils were either resting neutrophils activated *ex vivo* or neutrophils isolated from blood or tissue during a disease condition or experimental model. We have clarified this in the results section of the paragraphs “**Transcriptional conservation in resting neutrophils**”: “To systematically analyze which genes display similar and divergent expression across species, we integrated transcriptional profiles of resting (not activated) neutrophils available through the Sequence Read Archive (SRA). In this context, resting neutrophils were defined as those isolated from blood or tissue in the absence of disease or experimental manipulation.” and “**A core inflammation program is shared across conditions and conserved across species**”: “Here, resting neutrophils were defined as above and compared with their respective inflammatory condition.”

In Figure 3c, the log₂ fold change for upregulated and downregulated genes are shown for each individual study. Each column is one gene and each row is one comparison. Figure 3d shows the 179 core genes as a comparison between resting controls and activated cells for each sample individually. This view confirms that expression of these genes is mostly absent in resting neutrophils (both species) and that they become induced in the activated conditions.

We clarified this in the text as follows: “A total of 221 genes displayed consistent changes in inflammation across studies: 179 genes were upregulated across comparisons (the “core inflammation program”) and 42 genes were downregulated (Figure 3c). Effect sizes of those 221 up- and downregulated genes agreed well across all tested comparisons and across species (Figure 3c, Supplementary Figure 2b).”

4) Can the authors explain how they chose the two transcription factors JunB and Cebpbeta for their knock out model using the HoxB8 out of the top transcription factors with the highest predicted activity?

Response: When evaluating predicted regulatory activity and change in chromatin accessibility together, JunB emerged as most prominently affected transcription factor. In contrast, although the predicted regulatory activity of C/EBPB was high, its fold change in inflammation was smaller (Figure 5a). Furthermore, while a strong increase in chromatin

accessibility in inflammation was detectable for JunB, this was not the case for C/EBPB (Figure 5d). We therefore pursued the idea that the combination of functional evidence in neutrophil activation for these transcription factors and the different predicted magnitude of involvement in our conserved response analysis would make knockouts of JunB and C/EBPB ideal to evaluate their influence on the core inflammation genes.

Corresponding RNA-seq data from HoxB8 cell lines carrying these knockouts was available from the Udalova lab (PMID 34282331), so that their data could be integrated seamlessly into our analysis. We have added a section in the results to clarify this:

“When evaluating predicted conserved regulatory activity and change in chromatin accessibility together, JUNB emerged as a prominently affected transcription factor and has previously been shown to control neutrophil activation³⁶, and to be highly expressed upon neutrophil activation⁴³. On the other hand, CEBPB has previously been shown to be a key transcription factor mediating emergency granulopoiesis⁴⁴ and showed a high predicted regulatory activity in our analysis with limited changes in chromatin accessibility.”

5) Identification of a core program between mice and human does not mean that the pathophysiological mechanisms are similar between murine models and human diseases

The authors found a high conservation of lineage-specific gene, particularly the ones whose refers to neutrophils. According to the integrative analysis of transcriptomes, they could define a core inflammation program relatively well- preserved between mice and humans. Later they validated this program in different experimental models.

Although the authors highlight a restricted number of common and conserved molecular effectors between mice and human, the conclusion that the neutrophil activation would rely on similar pathogenic mechanisms is misleading. Most likely, the mechanisms that would be relevant for a given pathology would be outside of this core program which is, by definition not specific at all.

Although this analysis is useful and well done, the data that are presented should not be overinterpreted. The authors should clearly explain the intrinsic differences between murine and human neutrophils related to infectious diseases for example. The authors mention the lack of defensins in murine neutrophils but there are many differences due to the co-evolution of pathogens with the host in term of immune adaptation.

The authors could refer to a previously published papers. For instance the article by W. Nauseef in Immunol Review (DOI: 10.1111/imr.13154) has a complete review of the literature on this topic and clearly highlights the differences between murine and human neutrophils. In the conclusion, the authors explain how both human and murine systems could be complementary in neutrophil studies provided that some cautions are taken.

The discussion of the present manuscript could be enriched by these elements which should be taken into consideration for any researcher working on neutrophils.

Response: We agree that even if gene expression is highly conserved in resting neutrophils and a core inflammation program shared across both species exists, the molecular mechanisms can and will still be different given known species differences. We added an entire **new section to the discussion** to discuss this limitation of the study and also added

the reference suggested and several others discussing known differences between murine and human neutrophils: “In the context of neutrophils, fundamental differences between humans and mice exist^{59, 60}. Those differences must be considered when using the mouse as a model to study neutrophil function, especially in disease, as previously discussed⁶¹. Granule proteins found in neutrophils play a key role in defense against infection. An important difference in the granule protein repertoire includes α -defensins, which exercise antimicrobe^{62, 63} and chemotactic⁶⁴ activity and are absent in murine neutrophils. It is also known that murine neutrophils express less MPO, leading to a more limited capability to produce hypochlorous acid compared to their human counterpart⁶⁵. The importance of cytokine production by neutrophils has been increasingly recognized^{66, 67}, with some cytokines such as IFN- β and IL-17 apparently expressed in murine and not human neutrophils. The different immunoreceptor reservoir⁶⁸ is, in part, a result of pathogen responses that are exclusive to the human species. For example, human neutrophils express specific CEACAMs that mediate uptake of the human-specific pathogen *Neisseria gonorrhoea*⁶⁹, which must be taken into account when modeling neutrophil responses to this pathogen⁷⁰. Taken together, these studies provide important context to be taken into account when interpreting the core inflammation program identified.”

Reviewer #2 (Remarks to the Author):

Although the transcriptional profiles of neutrophils in normal and pathophysiological conditions in mice and humans have been demonstrated separately in recent studies, in context of comparing these profiles among mice and humans to identify the core features that are shared between these two species in different conditions is informative. The findings are interesting, and the paper is well-written.

However, in addition to demonstrating the general transcriptomic profiles and pathways analyzed, it would be interesting to have revealed the effector molecules (which promote inflammation, and tissue injury) of the neutrophils among the groups they included in their study. For example, ROS levels and NETosis.

Response: We thank the reviewer for their positive assessment of our study and their helpful suggestions for further improvement.

We now performed a focused analysis on the pathways presented in the Xie et al. paper, including effector molecules. The pathway “Chemotaxis (GO:0030593)” was consistently enriched in core inflammation genes in many samples. The gene set “ROS production (GO:1903409)” was enriched in the core inflammation genes in two studies. **(Reviewer only Figure 1).**

Aged neutrophil markers are CXCR4+ and CD62L (L-selectin) low/negative; however, in Supplementary Figure 7, flow cytometry analysis of neutrophil aging in vitro, there is no information regarding these markers.

Response: Neutrophil aging has typically been studied *in vivo* in mice and is not well understood how classical aging markers behave in an *in vitro* setting.

To address this concern, we performed new experiments and measured the expression of CXCR4, CD62L and CD101 in addition to core inflammation members in resting neutrophils at 0 hours and after 48 hours. The results are shown in **new Supplementary Figure 8.**

Surface expression of CXCR4 increased over time in human neutrophils, confirming our previous results (PMID 35168946). Expression of CD62L and CD101 decreased. In mouse neutrophils, CXCR4 expression decreased, while CD62L and CD101 increased, suggesting a more mature neutrophil phenotype, while the increase in CD62L suggests that any potential shedding by activation is replenished by additional production/externalization. It is therefore likely that the bone marrow neutrophils used in the experiments continue their maturation *in vitro*. As the neutrophils in this *in vitro* system do not resemble classically aged neutrophils *in vivo*, we have changed the term in the text accordingly: “Prolonged cell culture without activation led to an increase in CXCR4 and loss of CD62L and CD101 in human cells, while murine cells showed a reversed phenotype with upregulation of CD62L and CD101 as well as a downregulation of CXCR4, suggesting continued maturation of bone marrow neutrophils *in vitro* and not classical neutrophil aging (Supplementary Figure 8).”

What were the criteria for including and excluding the publicly available data sets that the authors analyzed in their manuscript? Per Fig 3A, the authors have mentioned the stimuli or disease conditions of the data they studied. Still, the question is why the authors included Lupus and JIA but not the bacterial infection models or the clinical samples of sepsis.

Response: We used Gene Expression Omnibus to search for RNA-seq datasets which contained both resting (not activated) neutrophils which could be used as internal control in addition to activated neutrophils from different inflammatory conditions (**Supplementary Figure 1**). Some studies identified in our initial search had to be excluded due to unavailability of controls or lack of data availability and it is possible that we may have missed additional studies of interest. We are now providing a more comprehensive overview of the sample selection in **Supplementary Figure 1**.

In this revised manuscript, we additionally included four recently published single cell RNA-seq studies. The inflammatory conditions encompassed bacterial infection in mice with *E. coli* (Xie et al. *Nature Immunology* 2020), patients with COVID-19 infection (Combes et al. *Nature* 2021) and stimulation with G-CSF, IFN- β or INF- γ (Montaldo et al. *Nature Immunology* 2022). All inflammatory settings clearly show an enrichment in the core inflammation program, confirming that the core inflammation program constitutes a selection of genes from which neutrophils preferentially upregulate expression when they become activated.

A typographical error: Fig 5D in the result's section statistics (0.79) and Fig 5D statistics (0.0011) are not matched well.

Response: We corrected the typographical error in Figure 5.

Reviewer #3 (Remarks to the Author):

Grieshaber-Bouyer and colleagues present a re-analysis of neutrophil transcriptomic datasets in both mice and human. The goal is to identify and characterize conserved core signatures both at baseline (resting) and after neutrophil activation following different pro-inflammatory stimuli.

The authors first identify lineage-defining genes and then build classes of genes based on the concordance of their expression in both species. Using a panel of existing studies, the authors perform differential expression between resting and activated neutrophils to

compile a core inflammatory signature (179 genes), arguably conserved across stimuli and species. Further data integration is performed using linear models and co-expression network analysis. Additional computational analyses are carried out to validate the parallelism between gene regulatory programs in human and mice using transcription factor enrichment and activity analysis, along with chromatin accessibility data from a specific model of acute inflammation. The regulome of 2 of the identified factors (JUNB and CEBPB) is computationally characterized from public expression data generated on ex-vivo neutrophils knock-out experiments. Flow cytometry is then applied to address how the core transcriptional inflammatory signature is recapitulated at the protein level, again revealing both conserved and species-specific features. Finally, the tissue-specificity of the core program is evaluated using stimulation experiments and flow cytometry for a small panel of selected genes.

In its current form, and although the re-analysis effort constitutes an interesting meta-analysis, it is this referee's opinion that the results and conclusions of this work largely rely on previous studies consisting on a small number of samples and limited diversity, while the scope of the study is definitely ambitious, comprising human, mouse and diverse pro-inflammatory signals. The identification of a conserved, universal baseline and inflammatory core transcriptional signature in neutrophils would likely require a properly designed dataset, and its comprehensive characterization and validation at the chromatin, protein and regulatory level should rely on a properly matched multi-omic screen.

In addition, a more detailed discussion, comparative analysis or integration with relevant studies reporting fine-grained neutrophil transcriptional signatures is lacking.

Response: We thank the reviewer for their honest criticism of our work. Although the suggestion of a multi-omic screen encompassing chromatin, protein and regulatory level is appealing for future studies, this is largely outside the scope of this present work. Nevertheless, we have been able to address the concerns raised through the inclusion of new data and new analyses and a more detailed discussion and hope the new data and demonstrated rigor in re-analysis will convince the reviewer that our study adds important information to the field, in addition to presenting an exciting approach to analyzing transcriptomic data across species.

My major concerns are summarized below, where I focus on analytical and statistical details and how they relate with the major conclusions drawn by the authors:

- Neutrophils are characterized by a relatively less complex transcriptome as compared to other immune subsets. Other factors specific to this population (high RNase content, cell loss during library preparation) are known to affect the quality of neutrophil transcriptomics. These factors can determine the suitability of some of the analytical choices but are not discussed by the authors. For instance, the higher correlation between mouse and human lineage-defining genes as compared to other immune subsets could be due a lower transcriptome complexity, reflected in a higher segregation between low and highly expressed genes in neutrophils. As lineage-defining genes are likely to be on the high-expression tail of the distribution, this could inflate the correlation across species for these genes.

Similarly, any randomization analysis (figure 5D) should account for this using matched overall expression levels between target and random sets. Otherwise, random sets might be

largely constituted by lowly expressed genes, artificially biasing the differential enrichment results.

Response: We are grateful for this comment and agree with this assessment.

In the results section describing Figure 1, we now acknowledge that technical factors may have influenced the relatively higher correlation in neutrophils compared to other lineages: “Of note, although these data indicate a higher correlation in neutrophils compared to other lineages, this effect may have been influenced by smaller library complexities in neutrophils.”

In our new randomization analysis, we now account for the expression abundance of core inflammation program genes by binning gene expression in each comparison into 25 bins and then sampling from the respective bins with matched overall expression. This change is reflected in the Methods section: “The control expression was calculated as previously described⁹⁴. For each comparison, the gene expression was distributed in 25 bins. Then, each core inflammation program member was assigned to its respective bin. The randomized sets were then sampled according to the distribution of core inflammation program gene expressions. This sampling was repeated 1000 times.” and is now implemented in **revised Figure 5e**.

- On a related note, the identification of lineage-defining genes is based on a very small number of neutrophil samples (3 for human, 6 for mouse). As no additional details about sequencing factors (e.g. total sequencing depth and library quality) are provided, a discussion or more in-depth evaluation of the robustness of the identified signatures would be required. And on a technical note, k-mer based RNA-Seq methodologies like the one used by the authors, have been reported to be less accurate for lowly expressed genes, potentially aggravating the effect of low-transcriptome complexity on the results. And being a transcriptome-based approach, further data loss can be expected due to a well-known prevalence of intron retention events in neutrophils. Unfortunately, “technical” factors cannot be avoided but need to be taken into account and acknowledged, particularly when they are specific to the cell type of interest.

Response: We agree that a more rigorous assessment of library quality is necessary.

Therefore, we amended our supplementary materials by adding a figure detailing sequencing factors of all included samples and revised the manuscript to include a more detailed discussion on their effects on our findings. More quality control metrics can be found in **Supplementary Table S1** and **Supplementary Figure 10**.

To further test the robustness of the identified program, we repeated our entire analysis using spliced STAR alignment to the respective reference genomes followed by quantification of transcripts with Salmon’s alignment-based quantification mode. The results obtained from the more exact STAR alignment and the faster, k-mer based Salmon pseudoalignment, are highly concordant.

Reviewer only Figures 2–5 show the comparison in gene expression between STAR (x axis) and Salmon (y axis) for each sample. **Reviewer only Figures 2–3** show the data used in lineage analysis and **Reviewer only Figures 4–5** show the analysis of resting and inflamed neutrophil samples.

Reviewer only Figures 6–8 show the lineage analysis and resulting core inflammation program based on STAR alignment. As the results are virtually identical to the results

generated using Salmon alignment (**Figures 1–3** of the main manuscript), we opted not to change the main figures.

Based on the STAR alignments, we also quantified intronic reads within each sample in our lineage dataset using RSeQC. This analysis confirmed the higher prevalence of intronic reads in murine neutrophils as compared to most other murine lineages but did not show a strong bias for human neutrophil samples to contain more reads mapped to intronic regions than other lineages (**Reviewer only Figure 9**). As the higher prevalence of intronic reads appears to be a result of the unique biology of neutrophils, it cannot serve as quality control measure in these cells.

- Similarly, most of the public data compiled and re-analyzed by the authors in the context of inflammatory responses is based on very small case-control studies (figure 3). I understand the argument to restrict the analysis to studies with both case and control data available. But the combination of the various factors (species+model+tissue+stimuli) added to the limited number of replicates per experiment severely limits the scope of what can be done. Again, no information about data quality factors is included (total depth/reads in each experiment?), and how this affects summary statistics. The authors make the correct choice on building summary statistics of dataset-specific differential expression results, but the universality of the reported results is doubtful in my opinion (Figure 3C), showing limited or inconsistent up-regulation of many “core” inflammatory genes across datasets. Supplementary Figure 2 is also concerning given the variability in the reported numbers of differentially expressed genes across different studies (even with similar stimuli), and the significance of the identified core. Examples are Supp. Figure 2B showing a significant number of core inflammation genes that are exclusively regulated in one species; the lack of reproducibility shown for the down-regulation of CD101 in Supp. Fig. 2C (with no effect size in most samples), similar to the aforementioned in the core program shown in Figure 3C; the relatively low and scattered GSEA enrichment scores in figure 3E across datasets.

Response: Supplementary Figure 2a shows a surprisingly strong coherence between different studies using the same stimuli. For example, GM-CSF stimulation lead to 359 DE genes in one study (Thomas et al. 2015) and to 339 DE genes in another study (Wright et al. 2013). Of these, 206 genes overlap (**Reviewer only Figure 10**). Importantly, 52 and 58 of these differentially expressed genes are part of the core inflammation program and 45 of those overlap.

The goal of our study was not to describe an exclusive list of genes which are part of the core inflammation program. We bring up the idea that a group of genes exists, from which neutrophils preferentially draw when they become activated. Given the influence that both the selection of studies as well as analytical strategies can have, we now discuss this limitation in the discussion as follows: “It is important to note that different analytical strategies may be used to derive this core inflammation program, each detecting a varying number of genes. The situation is similar for differential gene expression in general, which depends on the chosen method, as has been reviewed extensively⁵². Nevertheless, our analysis indicates that a group of genes exist from which neutrophils preferentially draw when they become activated across humans and mice and across a large range of conditions and disease states.”

- Regarding the attempted global analyses across studies, several methodological choices are not explained, and the results insufficiently interpreted:

o Why do the authors perform **linear modeling based on log₂(TPM+1) scaled data**? TPM is a within library normalization and therefore strongly affected by library-specific factors that may not be completely, or even partially corrected by a random term in the model. Did the authors **try batch-corrected data, or between-samples variance stabilization (e.g from DESeq)**? I'd say that a majority of core inflammatory genes are not identified as significant in Supp. Figure 3 (reported is 49 genes out of 179, but many other genes not in the core set show similar statistics, this needs to be discussed).

Response: In response to this suggestion, we adapted our preprocessing approach to the global analyses across studies as detailed in methods. We used `Combat_seq` to perform batch correction between individual studies, calculated TMM normalized and log₂-transformed counts per million which were subsequently quantile normalized. We then used those values to fit linear mixed models as described in the original manuscript.

The changes are explained in the Methods as follows:

“Linear Mixed-Effect Model

To validate the core inflammation program derived from Fisher's combined test, we globally tested for differential expression between resting and inflamed cells including all samples used in the Fisher's combined testing approach. We accounted for batch effects by correcting gene counts for study using `ComBat-Seq`¹⁰⁴ (sva v3.44.0). From batch-corrected counts, we calculated TMM-normalized log₂ counts per million that were subsequently quantile normalized and then used as input for linear modeling. Modeling was implemented using `lme4`⁸⁶ (v1.1-29) to fit a linear mixed-effects model (LMM) to normalized counts. The linear formulae we fit for each gene were defined as *full: expression ~ condition + 1|study* and *reduced: expression ~ 1|study*, where the variable to test for was *condition*, and the *study* was used as the covariate that was considered to be the random effect. We retrieved β as an estimate for the log₂(FC) from the full model and subsequently performed a likelihood ratio test to compare the *full* with the *reduced* model and to retrieve the respective *P*-values. *P*-values were then adjusted using the Benjamini-Hochberg procedure. We accounted for batch effects by correcting gene counts using `ComBat-Seq`¹⁰⁴ (sva v3.44.0)”

o WGCNA results highlights some of my concerns above if my interpretation is correct. From Supp. Figure 5, sample-specific effects are apparent from the stripes of highly regulated genes in inflammatory conditions. This is expected but it can definitely affect the analysis. No information about the concordance across replicates, which is not apparent from that figure, is provided. Modules 33 and 34 are virtually identical, and many genes in these modules are downregulated in mouse inflammatory neutrophils, with opposite trend for most genes in human neutrophils. Module 10 comprises two subsets of genes with up and down-regulation after inflammation, which questions the concept of co-expression within a given module.

Response: We re-ran WGCNA with an updated preprocessing strategy that was described above for linear mixed modeling across studies and a more stringent filter to remove lowly expressed genes. Specifically, we implemented a low count filter, applied quantile normalization between samples as recommended by the authors of WGCNA, corrected the network type to a signed network (this resulted in the split between modules 33 and 34 not occurring any more), and used a soft thresholding power:

“Gene expression modules using WGCNA

For WGCNA³⁹ analysis, we selected the same samples that were used for differential expression testing. We accounted for batch effects by correcting gene counts using ComBat-Seq¹⁰⁴ (sva v3.44.0). From batch-corrected counts, we calculated TMM-normalized log₂ counts per million that were subsequently quantile normalized and then used as input for WGCNA. The network was constructed as a signed network, using a soft thresholding power of 13, a minimum module size of 30, and a merge cut height of 0.25. Modules with more than 1000 genes were removed from subsequent analyses.”

Only upregulated co-regulated genes are included in this updated analysis. We additionally updated WGCNA parameters to reflect the adjusted preprocessing strategy and follow best practices as recommended by the WGCNA package authors. This resulted in a more consistent expression pattern of all member genes for modules that are enriched in inflammatory response genes, resulting in **updated Supplementary Figure 5**.

- Regulatory analyses:

- o Accessibility analysis: ATAC-Seq data is limited and was obtained from a very specific model of acute inflammation. How appropriate this data is to validate a signature obtained from a very heterogeneous, across-species dataset is uncertain and not discussed. In addition, the significance of the numbers provided in page 7 (lines 252-258) is unclear without a proper **evaluation of the overall, genome-wide changes in accessibility**. The authors seem to rely on the results reported in the original study to perform some intersection analysis. In general, it is my opinion that the scope and relevance of the ATAC-seq data is very limited.

Response: We agree on the importance of discussing the model used for the accessibility analysis and have included the following paragraph into the discussion: “While this model is very specific, it covered neutrophils from different maturation stages and presented the opportunity to study transmigrated and activated neutrophils separately. Further, analysis of the transcriptome on a single cell level in both *in vivo* and *in vitro* inflamed neutrophils of both species allowed us to validate the core inflammation program. While the overall enrichment of the proposed gene set on a pseudo-bulk level was clearly evident, our analyses also suggested significant heterogeneity within the population of inflamed neutrophils, consistent with recent analyses^{7, 9, 40, 41}. These analyses further highlight the predictive value of the program in a method not used in its generation.”

While the genome-wide changes in accessibility have been analyzed in detail by Khojraty et al. (Figure 2 in the original publication), we agree that the observed accessibility changes should be put in context. For the question of interest (do core inflammation genes show increased accessibility compared to other genes?), we thus performed 1000 repeats of expression-matched background gene selection and inferred the respective intersect sizes. This is now reflected in the section “ATAC-sequencing analysis” of the Methods, and results are depicted in Figure 5 (see legend, Figure 5B):

“ATAC-sequencing analysis

We retrieved ATAC-sequencing data from mice that were subjected to the air pouch model of acute inflammation (GEO: GSE161765, mapped to the GRCh38 genome). Genes annotated based on differentially accessible peaks as defined in the study ($P_{adj} < 0.05$, fold change > 1.5) were compared with the conserved upregulated genes as defined in the core inflammation program. The ratio and number of core inflammation program genes that were associated with projected increased accessibility served as an input for pairwise

Fisher's exact tests, *P*-values were adjusted using the Benjamini-Hochberg method. For each comparison, the significance of the number of core inflammation genes with increased accessibility was retrieved by comparing the number with the results of this analysis using a 1000-fold repeated random selection of expression-matched background genes (as described below for RNA-sequencing)."

Of note, the observed group size for the core inflammation program is significantly higher in all comparisons. Taken together, we see the observed changes in accessibility in combination with the motif enrichment analysis as complementary to the transcription factor enrichment analysis that was derived from RNA-seq. We agree on the importance of clarifying the scope of this validation and have included this idea in the discussion: "Furthermore, the ATAC-seq data from the air pouch model of inflammation and RNA-seq data from zymosan activated HoxB8 samples represent only select validation strategies in specific modalities of inflammation."

o Everything is focused on up-regulation in inflammatory conditions, which is understandable. Are there any repressive programs after activation/migration the authors can comment on?

Response: Khoyratty et al. report closing peaks in regions associated with chromatin DNA binding, the nuclear membrane, and the peroxisome (Figure 2d-e and supplementary Figure 3a in original study). Filtering peaks with decreasing accessibility for genes that were downregulated in the meta-analysis showed a weak trend (e.g., 11 of 42 genes associated with decreased accessibility in the air pouch versus blood comparison versus 0 of 42 genes associated with decreased accessibility in the blood versus bone marrow comparison) which was not statistically significant (**Reviewer only Figure 11**).

- Given the extensive literature on the transcriptional programs in neutrophils, and the numerous recent studies using single-cell approaches to unveil the heterogeneity of this cell type, I'd expect a detailed comparative analyses between the summary results obtained in this meta-study and existing knowledge in the field. Instead, the discussion is limited to an enumeration of the analyses and results of this work. I am left wondering about the novelty of the findings reported here, the relevance and novelty of the transcription factor analyses results, how the core program identified here relates with the observed heterogeneity of human and mouse neutrophils, or even how their protein marker results could be interpreted (or even integrated somehow) with existing single-cell multi-omic datasets (CITE-Seq). Unfortunately, the lack of an in-depth evaluation of any novel findings as compared to a recapitulation of existing knowledge adds to the limited scope of the computational results.

Response: In our revised manuscript, we included the additional analysis from four single-cell RNA-seq datasets from three studies (bacterial infection in mice with *E. coli*, Xie et al. *Nature Immunology* 2020; patients with COVID-19 infection, Combes et al. *Nature* 2021; stimulation with G-CSF, interferon or gamma-interferon, Montaldo et al. *Nature Immunology* 2022). All studies clearly show an enrichment in the core inflammation program in the activated neutrophils.

In addition, we re-wrote the discussion to put the findings of the core program into better context:

“Finally, we validated key components of the predicted core inflammation program experimentally. Using primary human and murine neutrophils, we showed that the surface proteins CD14, CD69, IL-4R, CD40 and PD-L1 are induced by *in vitro* cytokine stimulation, and this upregulation is observable in both species, although CD40 was restricted to a small subset of neutrophils in humans, as expected ⁴⁶. This finding further underlines the conserved character of the inflammation program as presented in this study. Interestingly, while neutrophils from different murine tissues upregulated the inflammatory response markers, the magnitude of upregulation differed across bone marrow, spleen and blood, suggesting that the tissue origin of neutrophils is an important consideration in experimental studies.

The upregulation of IL-4R we observed is concordant with reports of IL-4R upregulation during sterile information in mice, with implications for diseases that are IL-4 mediated⁵³. CD14 has recently been shown to be an important, highly cell-specific mediator of TNF response in a murine sepsis model ⁵⁴. Interestingly, CD14⁺ macrophages and neutrophils were found to be key players leading to lethality in response to TNF (with improved survival in CD14-deficient mice), which provides a model for the cytokine storm seen in severe sepsis and provides evidence for the complexity of CD14-mediated inflammatory response beyond TLR-signaling. These examples highlight the importance of core inflammation program members and stress the need to study them in a broad variety of inflammatory contexts.”

In addition, we updated the limitations of the study:

“The derivation of the core inflammation program was limited to bulk RNA-sequencing samples, since a similar analysis using single-cell studies requires datasets that are only now beginning to emerge. To circumvent potential batch effects, we focused our analysis on studies with internal controls of resting neutrophils, excluding other potentially interesting studies containing only neutrophils harvested from inflamed sites. Analyzed samples were also limited by technical factors, including the known intron retention in neutrophils ⁷¹, as well as the less complex transcriptome associated with low RNA and high RNase content. Furthermore, analysis of single cell RNA sequencing data, the ATAC-seq data from the air pouch model of inflammation and RNA-seq data from zymosan activated HoxB8 samples represent only selected validation strategies in specific modalities of inflammation, which might limit the generalizability of some of the findings.”

REVIEWERS' COMMENTS

Reviewer #1 (Remarks to the Author):

The revised version of the manuscript by Hackert et al has been improved in term of clarity and relevance for the analysis of disease mechanisms using murine models. Basically, the authors were very responsive and have made some efforts to explain the methods that they have used to draw their conclusions.

Now, the authors have included more discussion in their conclusion to emphasize that the differences between the murine versus human neutrophils.

Reviewer #2 (Remarks to the Author):

The authors have adequately addressed my previous comments. I have an additional minor comment that I believe can be addressed through further discussion in their manuscript.

The authors have effectively demonstrated that prolonged cell culture without activation leads to an increase in CXCR4 expression and a concurrent loss of CD62L and CD101 in human cells (aged phenotype), as depicted in Supplementary Figure 10. Within human neutrophils, the upregulation of CD40 (antigen-presenting marker) was found to be confined to a minor subset, accounting for approximately 2% of neutrophils, as illustrated in Figure 6C.

In a recent study, a distinct subset of neutrophils termed "antigen-presenting aged neutrophils" (APANs) was identified (Jin H, et al, J Clin Invest 2023; 133:e164585). This unique neutrophil subpopulation exhibits a dual phenotype of antigen presentation and cellular aging, driving a functionally pro-inflammatory and hyperNETotic response. An intriguing avenue for discussion in their paper would be the potential presence and significance of APANs within both human and mouse neutrophil populations. This consideration could enrich the current findings and extend the implications of the study's outcomes.

Reviewer #3 (Remarks to the Author):

The authors have provided a satisfactory response to my original remarks, including extensive re-analysis, additional results and clarifications, and have now appropriately acknowledged some of the limitations of their study. Although the scope and cross-species universality of the core program identified here may remain debatable, it is this referee's opinion that this manuscript is a relevant contribution to the field.

A final minor comment: I encourage the authors to double-check the numbering and reference to the various figures and materials throughout the paper, which in the format I had available for review was not easy to track.

REVIEWERS' COMMENTS

Reviewer #1 (Remarks to the Author):

The revised version of the manuscript by Hackert et al has been improved in term of clarity and relevance for the analysis of disease mechanisms using murine models. Basically, the authors were very responsive and have made some efforts to explain the methods that they have used to draw their conclusions.

Now, the authors have included more discussion in their conclusion to emphasize that the differences between the murine versus human neutrophils.

Response: We thank the reviewer for their positive assessment of the revised manuscript as well as for the constructive feedback that has contributed to the improvement of our manuscript.

Reviewer #2 (Remarks to the Author):

The authors have adequately addressed my previous comments. I have an additional minor comment that I believe can be addressed through further discussion in their manuscript.

The authors have effectively demonstrated that prolonged cell culture without activation leads to an increase in CXCR4 expression and a concurrent loss of CD62L and CD101 in human cells (aged phenotype), as depicted in Supplementary Figure 10. Within human neutrophils, the upregulation of CD40 (antigen-presenting marker) was found to be confined to a minor subset, accounting for approximately 2% of neutrophils, as illustrated in Figure 6C.

In a recent study, a distinct subset of neutrophils termed "antigen-presenting aged neutrophils" (APANs) was identified (Jin H, et al, J Clin Invest 2023; 133:e164585). This unique neutrophil subpopulation exhibits a dual phenotype of antigen presentation and cellular aging, driving a functionally pro-inflammatory and hyperNETotic response. An intriguing avenue for discussion in their paper would be the potential presence and significance of APANs within both human and mouse neutrophil populations. This consideration could enrich the current findings and extend the implications of the study's outcomes.

Response: We appreciate the reviewer's feedback and the additional comment. We have carefully considered the suggestion and agree that discussing the potential presence and significance of "antigen-presenting aged neutrophils" (APANs) in both human and mouse neutrophil populations would be a valuable addition to our manuscript.

In response to the suggestion, we have included a new paragraph that draws a connection between our findings and the recent study by Jin et al.:

"Recently, Jin et al. identified a distinct neutrophil population termed "antigen-presenting aged neutrophils (APANs)"⁵³. In humans, this population was characterized as CD66b⁺CXCR4⁺CD62L^{lo}CD40⁺CD86⁺, while in mice, they were identified as Ly6G⁺CXCR4⁺CD62L⁻/^{lo}MHCII⁺CD40⁺CD86⁺. APANs were capable of inducing CD4 T cell proliferation via IL-12 and exhibited a hyper-NETosis phenotype. The presence of these neutrophils in patients with sepsis was associated with increased mortality. While we also observed the upregulation of key marker genes like *CD40* in our study's core inflammation program, APANs displayed distinct features, such as elevated levels of *CXCR4* and coexpression with *CD74*, suggesting a unique

neutrophil polarization state discriminable from both neutrophil aging and canonical activation. The phenotype observed by the authors suggests the importance of further studying APANs, their features and their role in antigen presentation in humans and mice.”

Reviewer #3 (Remarks to the Author):

The authors have provided a satisfactory response to my original remarks, including extensive re-analysis, additional results and clarifications, and have now appropriately acknowledged some of the limitations of their study. Although the scope and cross-species universality of the core program identified here may remain debatable, it is this referee's opinion that this manuscript is a relevant contribution to the field.

A final minor comment: I encourage the authors to double-check the numbering and reference to the various figures and materials throughout the paper, which in the format I had available for review was not easy to track.

Response: We are grateful for the reviewer's assessment of our revised manuscript. As suggested, we have carefully confirmed all references within the manuscript. We thank the reviewer for the valuable feedback which contributed to the quality of our work.